# On Improving Neurosymbolic Learning by Exploiting the Representation Space

## Abstract

We study the problem of learning neural classifiers in a neurosymbolic setting where the hidden gold labels of input instances must satisfy a logical formula. Learning in this setting proceeds by first computing (a subset of) the possible combinations of labels that satisfy the formula and then computing a loss using those combinations and the classifiers' scores. However, the space of label combinations can grow exponentially, making learning difficult. We propose the first technique that prunes this space by exploiting the intuition that instances with similar latent representations are likely to share the same label. While this intuition has been widely used in weakly supervised learning, its application in our setting is challenging due to label dependencies imposed by logical constraints. We formulate the pruning process as an integer linear program that discards inconsistent label combinations while respecting logical structure. Our approach is orthogonal to existing training algorithms and can be seamlessly integrated with them. Experiments on three state-of-the-art neurosymbolic engines, Scallop, Dolphin, and ISED, demonstrate up to 74% accuracy gains across diverse tasks, highlighting the effectiveness of leveraging the representation space in neurosymbolic learning.

## 1 Introduction

**Motivation.** *Neurosymbolic learning* (NSL), i.e., the integration of symbolic with neural mechanisms for inference and learning, has been proposed as the remedy for some of the most vulnerable aspects of deep networks Feldstein et al. (2024). Recent works have shown that NSL holds immense promise, offering, in addition, the means to train neural networks using weak labels Feldstein et al. (2023); Wang et al. (2023). We study the problem of learning neural classifiers in frameworks where a symbolic component "sits" on top of one or more neural classifiers and learning is weakly supervised Manhaeve et al. (2021). An example of our setting, referred to as NeSy, is presented below.

**Example 1.1** (NeSy example). *Consider a classical example of* NeSy*: learning an MNIST classifier $f$ using training samples of the form $(\{x_1, x_2\}, \phi)$, where $x_1$ and $x_2$ are MNIST digits and $\phi$ is a logical sentence that the gold labels of $x_1$ and $x_2$, $l_1$ and $l_2$, should satisfy Manhaeve et al. (2021). Unlike supervised learning, $l_1$ and $l_2$ are unknown to the learner. The logical sentence $\phi$ restricts the space of labels that can be assigned to $x_1$ and $x_2$. For example, consider the training sample $(\{x_1, x_2\}, \phi_1 := l_1 + l_2 = 8)$. According to this sample, any combination of $l_1$ and $l_2$ whose sum is 8 is valid and all other combinations are invalid, e.g., $l_1 = 2$ and $l_2 = 6$ is valid, but $l_1 = 3$ and $l_2 = 6$ is invalid. In total, there are 9 different combinations of $l_1$ and $l_2$ that satisfy $\phi_1$. The gold labels of $x_1$ and $x_2$ are 1 and 7, respectively. However, they are unknown during learning.*

NeSy is one of the most popular frameworks in the NSL literature, with DeepProbLog Manhaeve et al. (2021), NeuroLog Tsamoura et al. (2021), Scallop Huang et al. (2021a), Dolphin Naik et al. (2025), and ISED Solko-Breslin et al. (2024) being only a few of the frameworks that rely on it. In addition, as discussed in Wang et al. (2023), NeSy encompasses *partial label learning* (PLL) Cour et al. (2011); Cabannes et al. (2020), where each input instance is associated with a set of mutually exclusive candidate labels, and learning classifiers subject to constraints on their outputs, Steinhardt & Liang (2015); Zhang et al. (2020). and has wide range of applications, including fine-tuning large language models Li et al. (2024), aligning video to text Huang et al. (2024a), visual question answering Huang et al. (2021a), and learning knowledge graph embeddings Maene & Tsamoura (2025).

**Limitations.** Learning in NESY proceeds by first computing (a subset of) the possible combinations of labels that lead to the given learning target, subject to the symbolic component, and then computing a loss using those combinations and the classifiers' scores. However, learning becomes more challenging as the space of possible label combinations increases in size Marconato et al. (2023); Tsamoura et al. (2025) . This is because supervision becomes weaker. The question arises: *Are there circumstances where we can safely discard specific label combinations?*

**Contributions.** We are the first to propose a plug-and-play technique to reduce the space of candidate label combinations by *exploiting the inconsistency between the representation space and the space of candidate label combinations* in general-purpose neurosymbolic frameworks. Our intuition is that if the latent representations of two instances are very close, then they belong to the same class, and hence share the same gold labels. When applied to NESY, this intuition can substantially reduce candidate label combinations during training:

**Example 1.2.** *[Contd Example 1.1] Consider a second training sample* $(\{x_1', x_2'\}, \phi_2 := l_1' + l_2' = 2)$, *where* $l_1'$ *and* $l_2'$ *correspond to the gold labels of* $x_1'$ *and* $x_2'$. *According to this training sample, the valid combinations of labels for* $(x_1', x_2')$ *are* $(0, 2)$, $(1, 1)$, *and* $(2, 0)$. *If the latent representations of* $x_1$ *and* $x_1'$ *are very close, then* $l_1$ *must range in* $\{0, 1, 2\}$. *Hence, the number of candidate label combinations associated with the first training sample reduces from* 9 *to* 3.

Unlike NESY, the PLL literature has extensively investigated techniques that exploit the representation space to discard erroneous candidate labels during training Wu et al. (2022); Xia et al. (2023); Wang et al. (2022); Xu et al. (2021). In fact, the intuition in the above example has been successfully adopted in weakly supervised learning He et al. (2024). However, its straightforward adoption in NESY is problematic, as it can result in training samples associated with zero supervision, i.e., without candidate combinations of labels. To address this issue, we organize the training samples and their associated candidate label combinations into a graph, called the *proximity graph*. The edges in the graph reflect the proximity of instances in the representation space. Then, by generalizing the intuition in our example, we introduce the problem of discarding the maximum number of candidate label combinations subject to the edges in the graph under the constraint that each training sample is associated with at least one candidate label combination. We then propose a solution to this problem by casting it into an *integer linear program* (ILP) Srikumar & Roth (2023).

Our approach offers two unique benefits. First, it is complementary to NESY training algorithms: Our technique first discards candidate label combinations; then training proceeds with the remaining label combinations. Second, it can be employed in a training-free manner, i.e., we can discard candidate label combinations using a pre-trained encoder, such as a large vision and language model Li et al. (2023a), or ResNet He et al. (2015), before training. Alternatively, it may be applied during training, i.e., by using the encoder trained so far to extract features for the corresponding training instances, then training with the label combinations that have not been discarded, and repeating the process.

We evaluate the benefits of our technique, called CLIPPER, applying it in combination with three state-of-the-art neurosymbolic engines, SCALLOP, DOLPHIN, and ISED, on a variety of benchmarks that range from digit classification – the classic SUM-$M$, MAX-$M$, and HWF-$M$ benchmarks Manhaeve et al. (2021) – to visual question answering and video-to-text alignment. CLIPPER consistently improves the accuracy across all engines and benchmarks. In our most challenging benchmark, MUGEN, the baseline accuracy improves from 33.8% to 83.7%. The integration of CLIPPER with the above engines was rather straightforward: we employed CLIPPER to filter out pre-images during the pre-image computation phase and then used the remaining pre-images to train the classifier. Our main contributions are:

- We formalize the problem of discarding label combinations for a set of NESY training samples based on the proximity of the latent representations of their instances.
- We propose an ILP algorithm that guarantees that each training sample retains at least one candidate label combination while maximizing the number of discarded label combinations.
- We evaluate our technique with different neurosymbolic engines on a variety of benchmarks and demonstrate improvements in classification accuracy of up to 74%.

## 2 PRELIMINARIES

**Supervised learning.** For an integer $n \geq 1$, let $[n] := \{1, \ldots, n\}$. Let also $\mathcal{X}$ be the instance space and $\mathcal{Y} = [c]$ be the output space. We use $x, y$ to denote elements in $\mathcal{X}$ and $\mathcal{Y}$. We consider *scoring functions* of the form $f : \mathcal{X} \mapsto \Delta_c$, where $\Delta_c$ is the space of probability distributions over $\mathcal{Y}$, e.g., $f$ outputs the softmax probabilities (or *scores*) of a neural classifier. We use $f^j(x)$ to denote the score of $f(x)$ for class $j \in \mathcal{Y}$. A scoring function $f$ induces a *classifier* $[f] : \mathcal{X} \mapsto \mathcal{Y}$, whose *prediction* on $x$ is given by $\operatorname{argmax}_{j \in [c]} f^j(x)$. Supervised learning aims to learn $f$ using samples of the form $(x, y)$.

**Neurosymbolic learning.** We assume familiarity with basic notions of logic, such as the notions of variables, constants, predicates, facts, rules, and sentences. We use small for constants and predicates, and capitals for variables. We point readers that wish to learn this background to Li et al. (2023c). To ease the presentation, we assume a single classifier $f : \mathcal{X} \to \mathcal{Y}$. Notice, though, that our results straightforwardly extend to settings with multiple classifiers. Let $\mathcal{K}$ be a background logical theory. As mentioned in Section 1, $\mathcal{K}$ "sits" on top of $f$, i.e., it reasons over the predictions of $f$. Of course, this is possible by translating neural predictions into facts, e.g., returning to Example 1.1, DeepProbLog, SCALLOP, and DOLPHIN, create one fact of the form $digit(d, x_1)$ for each possible digit $d$ and associate this fact with the softmax score of class $d$ for $x_1$ (and similarly for $x_2$). Then, reasoning over those facts using $\mathcal{K}$ produces the overall outputs. Different frameworks may employ different reasoning semantics at testing time which is orthogonal to this work.

Unlike supervised learning, in NESY, each training sample is of the form $(\mathbf{x}, \phi)$, where $\mathbf{x}$ is a set of elements from $\mathcal{X}$ and $\phi$ is a logical sentence (or a single target fact in the simplest scenario). The gold labels of the input instances are unknown to the learner. Instead, we only know that the gold labels of the elements in $\mathbf{x}$ satisfy the logical sentence $\phi$ subject to $\mathcal{K}$. In Example 1.1, $\mathcal{K}$ is empty. However, in one of the benchmarks that we consider in our experiments, namely VQAR Huang et al. (2021a), $\mathcal{K}$ is commonsense knowledge from CRIC Gao et al. (2019).

The above may seem prohibitive for learning. However, $\phi$ and $\mathcal{K}$ allow us to "guess" what the gold labels of the elements in $\mathbf{x}$ might be so that $\phi$ is logically satisfied subject to $\mathcal{K}$. This is essentially the process of *abduction* Tsamoura et al. (2021). To align with the terminology in Wang et al. (2023), for a training sample $(\mathbf{x}, \phi)$, we use the term *pre-image*[1] to denote a combination of labels of the elements in $\mathbf{x}$, such that $\phi$ is logically satisfied subject to $\mathcal{K}$. The gold pre-image is the one mapping each instance to its gold label. By construction, each NESY training sample includes the gold pre-image. More details on abduction are in Tsamoura et al. (2021). Abduction allows us to "get rid of" $\phi$ and $\mathcal{K}$ and represent each training sample via $\mathbf{x}$ and its corresponding pre-images, i.e., as $(\mathbf{x}, \{\sigma_i\}_{i=1}^{\omega})$, where each pre-image $\sigma_i$ is a mapping from $\mathbf{x}$ into $\mathcal{Y}$. We use $\mathcal{D} = \{(\mathbf{x}_\ell, \{\sigma_{\ell,i}\}_{i=1}^{\omega_\ell})\}_{\ell=1}^{n}$ to denote a set of $n$ NESY training samples.

**Example 2.1.** *[Contd Example 1.2] Candidate pre-images for the first sample are:* $\sigma_{1,1} = \{x_1 \mapsto 0, x_2 \mapsto 8\}$, $\sigma_{1,2} = \{x_1 \mapsto 1, x_2 \mapsto 7\}$, *and* $\sigma_{1,3} = \{x_1 \mapsto 8, x_2 \mapsto 0\}$. *Two candidate pre-images of the second sample are:* $\sigma_{2,1} = \{x_1' \mapsto 0, x_2' \mapsto 2\}$ *and* $\sigma_{2,2} = \{x_1' \mapsto 1, x_2' \mapsto 1\}$.

Our notation of pre-images is equivalent to the notation of training samples in Wang et al. (2023). The only thing left to discuss is what is the learning objective in NESY. Each NESY framework adopts its own learning objective. For example, in DeepProbLog and SCALLOP, the aim is to minimize semantic loss Xu et al. (2018) or its approximations Huang et al. (2021a). The authors in Wang et al. (2023) formalize learning via minimizing *zero-one partial loss*, that is the probability $\phi$ not being logically satisfied subject to $\mathcal{K}$. Our work is orthogonal to the actual loss used for training. The notation used throughout our work is summarized in Table 6 in the appendix.

## 3 DISCARDING PRE-IMAGES BASED ON LATENT REPRESENTATIONS

We aim to reduce the number of candidate pre-images of the NESY training samples by exploiting inconsistencies with the representation space. The question naturally arises: *Can a reduction in the number of pre-images per training sample lead to classifiers with higher accuracy?* The NSL community has verified this claim both experimentally Tsamoura et al. (2021); Huang et al. (2021a) and theoretically Marconato et al. (2023); Tsamoura et al. (2025). For example, Marconato et al.

---

[1]Pre-images correspond to *proofs* in Tsamoura et al. (2021); Huang et al. (2021a); Manhaeve et al. (2021).

(2023) showed that the number of deterministic classifiers that minimize semantic loss Xu et al. (2018) is directly proportional to the number of abductive proofs per training sample (i.e., pre-images in our terminology), while Tsamoura et al. (2025) showed that the probability a classifier misclassifies instances of the given class is a direct function of the number of pre-images.

Central to our technique are two notions: *proximity graphs* and *consistency*. Proximity graphs are graphs whose edges reflect the proximity of latent instance representations. As we will see later, proximity between instances imposes restrictions on the pre-images. Consistency reflects whether a given pre-image abides by those restrictions. This section is organized as follows. Section 3.1 introduces our key notions and our new problem formulation. Section 3.2 presents our technique and provides optimality guarantees. Section 3.3 discusses variations of our formulation from Section 3.2.

### 3.1 NOTIONS AND PROBLEM STATEMENT

We start by introducing the notion of a proximity graph. Let $h$ be an encoder from $\mathcal{X}$ to $\mathbb{R}^m$.

**Definition 3.1** (Proximity graphs). *A proximity graph $\mathcal{G}_{\mathcal{D}}^h$ for $\mathcal{D}$ subject to $h$ is a directed graph that includes one node $(\ell, x)$, for each $\ell \in [n]$ and $x \in \mathbf{x}_\ell$, and, optionally, a directed edge from node $(\ell, x)$ to node $(\ell', x')$ if $h(x')$ is close to $h(x)$, for $x, x' \in \mathcal{X}$.*

The edges of the graph $\mathcal{G}_{\mathcal{D}}^h$ define proximity in the representation space. Notice that Definition 3.1 does not depend on either the encoder $h$ that will give us the latent representations, e.g., the encoder can be a pre-trained large vision and language model such as BLIP-2 Li et al. (2023a), or on the measure used to decide the distance in the representation space. We deliberately kept the vague term "close" in Definition 3.1 to support any distance measure a user may prefer. For example, an option is to define a distance threshold $\theta$ and add edges only between instances whose latent representations are less than $\theta$ apart. A second option is to add a directed edge $(\ell, x) \rightarrow (\ell', x')$ only if $h(x')$ is in the top-$k$ neighborhood of $h(x)$, for $x, x' \in \mathcal{X}$ – the use of directed edges gives us greater flexibility to adopt such definitions. Of course, the "better" the encoder $h$ is, the more effective our algorithm will be in pruning the non-gold pre-images.

The graph $\mathcal{G}_{\mathcal{D}}^h$ tells us when two instances of different samples are very close in the representation space. When two instances are very close in the representation space, they should be of the same class, sharing the same gold labels. Due to the dependencies among different labels in the pre-images, some candidate pre-images may satisfy the restriction that the corresponding instances should share the same gold labels. Others may not. The notion of *consistency* formalizes the above intuition.

**Definition 3.2** (Consistency). *For a proximity graph $\mathcal{G}_{\mathcal{D}}^h$, a pre-image $\sigma_{\ell,i}$ in $\mathcal{D}$ is consistent with an edge $(\ell, x) \rightarrow (\ell', x')$ in $\mathcal{G}_{\mathcal{D}}^h$ if there exists a pre-image $\sigma_{\ell',i'}$ in $\mathcal{D}$, such that $\sigma_{\ell,i}(x) = \sigma_{\ell',i'}(x')$ holds; otherwise, we say that $\sigma_{\ell,i}$ is inconsistent with $(\ell, x) \rightarrow (\ell', x')$. The pre-image $\sigma_{\ell,i}$ is globally consistent in $\mathcal{G}_{\mathcal{D}}^h$ if there does not exist an edge $(\ell, x) \rightarrow (\ell', x')$ in $\mathcal{G}_{\mathcal{D}}^h$ with which $\sigma_{\ell,i}$ is inconsistent.*

We present an example of Definition 3.2.

**Example 3.3** (Contd Example 2.1). *Assume the proximity graph for the two training samples in our running example includes edge $e_1 := (1, x_1) \rightarrow (2, x_1')$. Since there does not exist a pre-image associated with the second training sample mapping $x_1'$ to 8, the pre-image $\sigma_{1,3} = \{x_1 \mapsto 8, x_2 \mapsto 0\}$ is inconsistent with $e_1$. In contrast, the pre-image $\sigma_{1,1} = \{x_1 \mapsto 0, x_2 \mapsto 8\}$ is consistent with $e_1$, due to the existence of the pre-image $\sigma_{2,1} = \{x_1' \mapsto 0, x_2' \mapsto 2\}$. Generalizing this example, all the pre-images in the first training sample that map $x_1$ to a digit greater than 2 are inconsistent with $e_1$. The remaining pre-images are consistent with $e_1$. Now, consider the edge $e_1' := (2, x_1') \rightarrow (1, x_1)$. In the absence of other edges, all pre-images of the second training sample are globally consistent.*

Inconsistencies between pre-images and edges indicate violations of the restriction that the corresponding instances belong to the same class as we have seen in our running example. Hence, the corresponding pre-images need to be discarded. Definition 3.4 summarizes the process of discarding pre-images from a set of NESY samples based on such inconsistencies.

**Definition 3.4** (Pruning). *The pruning $\Pi(\mathcal{G}_{\mathcal{D}}^h)$ of $\mathcal{D}$ subject to $\mathcal{G}_{\mathcal{D}}^h$ is the set of NESY samples that results after removing from each training sample in $\mathcal{D}$ each pre-image that is inconsistent with an edge in $\mathcal{G}_{\mathcal{D}}^h$. The pruning is sound if at least one pre-image is preserved for each sample.*

---

**Algorithm 1** CLIPPER

---

**Inputs:** Encoder $h$; NESY dataset $\mathcal{D} = \{(\mathbf{x}_\ell, \{\sigma_{\ell,i}\}_{i=1}^{\omega_\ell})\}_{\ell=1}^n$.
**Outputs:** Pruned NESY dataset $\mathcal{D}'$.
$\mathcal{D}' := \varnothing$
**for each** mini-batch $\mathbf{b}$ of $\mathcal{D}$ **do**
    **find** the proximity graph $\mathcal{G}_\mathbf{b}^h$ for $\mathbf{b}$ maximizing (1).
    **for each** $\ell \in [n]$ **do**
        $\Omega_\ell := \varnothing$
        **for each** $i \in [\omega_\ell]$ **do**
            **add** $\sigma_{\ell,i}$ to $\Omega_\ell$ if $I'_{\ell,i} = 0$ in the optimal solution to (1).
        **add** $(\mathbf{x}_\ell, \Omega_\ell)$ to $\mathcal{D}'$
**return** $\mathcal{D}'$

---

Different proximity graphs have different edges. Hence, they may result in different prunings. A *gold proximity graph* for $\mathcal{D}$, denoted by $\mathcal{G}_\mathcal{D}^*$, is a graph that includes a directed edge from $(\ell, x)$ to $(\ell', x')$ if $x$ and $x'$ belong to the same class, where $\ell, \ell' \in [n]$, $x \in \mathbf{x}_\ell$, and $x' \in \mathbf{x}'_\ell$. We have:

**Proposition 3.5.** *For each $\ell \in [n]$, the $\ell$-th training sample in $\Pi(\mathcal{G}_\mathcal{D}^*)$ includes the gold pre-image.*

Due to Proposition 3.5, one might think that a strategy for discarding pre-images from $\mathcal{D}$ would be the following: (1) Construct a proximity graph $\mathcal{G}_\mathcal{D}^h$ including as many edges as possible[2]; and (2) Remove each pre-image that is inconsistent with an edge in $\mathcal{G}_\mathcal{D}^h$. *Does the above approach result in a sound pruning?* No, as we demonstrate in the example below:

**Example 3.6** (Contd Example 3.3)**.** *Consider also a third training sample $(\{x''_1, x''_2\}, \phi_3 := l''_1 + l''_2 = 16)$ and the edge $e_2 := (1, x_1) \to (3, x''_1)$. In the pruning of the proximity graph that includes both $e_1$ (see Example 3.3) and $e_2$, the first training sample will be associated with zero pre-images. This is because $x_1$ cannot range simultaneously in the domains $\{0, 1, 2\}$ and $\{7, 8, 9\}$.*

Cases such as those described in Example 3.6 are met when the encoder maps instances of difference classes very close in the representation space. In other words, while adding as many edges as possible to $\mathcal{G}_\mathcal{D}^*$ does not affect the soundness of $\Pi(\mathcal{G}_\mathcal{D}^*)$, this property does not hold in the general case.

To summarize the discussion so far, discarding pre-images from a set of NESY training samples reduces to finding a proximity graph whose edges reflect proximity in the representation space, according to Definition 3.1. However, we need to be careful on how we choose this proximity graph: too few edges may result in discarding very few pre-images; too many edges may result to prunings that are not sound, see Definition 3.4. The above gives rise to the following optimization problem.

**Problem 3.7.** *For an encoder $h$, find the proximity graph $\mathcal{G}_\mathcal{D}^h$ that leads to the pruning of $\mathcal{D}$ that (1) is sound, (2) includes all globally consistent pre-images, and (3) has the lowest total number of pre-images across all training samples.*

According to Problem 3.7, the desired proximity graph should maximize the number of discarded pre-images. Soundness ensures that we still have at least one pre-image in each training sample and, hence, we can use those samples for training. This assumption comes from the fact that, by definition, each NESY training sample includes the gold pre-image. Finally, we require the pruning of $\mathcal{D}$ to include all globally consistent pre-images as we have no evidence to discard these pre-images. In the next section, we cast Problem 3.7 as an ILP.

## 3.2 A LINEAR PROGRAMMING FORMULATION

To formalize Problem 3.7 as an ILP, we need to define the binary variables. First, we add a binary variable $E_{\ell,\ell',x,x'}$ for each $\ell, \ell' \in [n]$, $x \in \mathbf{x}_\ell$, and $x' \in \mathbf{x}'_\ell$, if $h(x')$ is close to $h(x) - h$ and "closeness" is an implementation choice as discussed in Section 3.1. The variable $E_{\ell,\ell',x,x'}$ is one if the resulting proximity graph includes the corresponding edge and zero otherwise. Second, we add a binary variable $I_{\ell,i}$ that corresponds to $\sigma_{\ell,i}$, that is the $i$-th pre-image of the $\ell$-th training sample,

---

[2]Under the assumption that the corresponding instances $x, x'$ are in fact close under $h$ and the distance measures in use, where $x, x' \in \mathcal{X}$.

for $\ell \in [n]$ and $i \in [\omega_\ell]$. The variable $I_{\ell,i}$ is one if $\sigma_{\ell,i}$ is in the pruning of $\mathcal{D}$ subject to the resulting proximity graph; otherwise it is zero. Finally, we add a binary variable $I'_{\ell,i}$ for each $\ell \in [n]$ and $i \in [\omega_\ell]$ that is the complement of $I_{\ell,i}$, i.e., it is one when $I_{\ell,i}$ is zero and vice versa. We are now ready to discuss the constraints of the linear program.

The first constraint is $I_{\ell,i} + I'_{\ell,i} = 1$ and states that the two variables are mutually exclusive. The second constraint is $\sum_{i=1}^{[\omega_\ell]} I_{\ell,i} \geq 1$, for each $\ell \in [n]$, and states that each training sample must include at least one pre-image. The third constraint is $I_{\ell,i} = 1$, for each $\ell \in [n]$ and $i \in [\omega_\ell]$, if $\sigma_{\ell,i}$ is globally consistent in any proximity graph that can be computed for the given training samples $\mathcal{D}$ subject to $h$ and the distance measures selected, see Definition 3.2. This constraint ensures that those pre-images will not be discarded in the pruning. The fourth constraint is $1 - E_{\ell,\ell',x,x'} + 1 - I_{\ell,i} = 1$ and expresses that $\sigma_{\ell,i}$ is inconsistent with the edge $(\ell, x) \rightarrow (\ell', x')$, see Definition 3.2. The remaining constraints define the domain. The objective is to maximize the number of discarded pre-images.

$$
\begin{aligned}
\textbf{objective} \quad & \max \sum_{\ell \in [n], i \in [\omega_\ell]} I'_{\ell,i}, \\
& \quad I_{\ell,i} + I'_{\ell,i} = 1, && \forall \ell \in [n], \forall i \in [\omega_\ell] \\
& \quad \sum_{i=1}^{[\omega_\ell]} I_{\ell,i} \geq 1, && \forall \ell \in [n] \\
& \quad I_{\ell,i} = 1, && \forall \ell \in [n], \forall i \in [\omega_\ell], \text{s.t.} \\
& && \sigma_{\ell,i} \text{ is always globally consistent.} \\
\textbf{s.t.} \quad & 1 - E_{\ell,\ell',x,x'} + 1 - I_{\ell,i} = 1, && \forall \ell \in [n], \forall \ell' \in [n], \forall x \in \mathbf{x}_\ell, \forall x' \in \mathbf{x}_{\ell'}, \text{s.t.} \\
& && \sigma_{\ell,i} \text{ is inconsistent with } (\ell, x) \rightarrow (\ell', x'). \\
& \quad E_{\ell,\ell',x,x'} \in \{0,1\}, && \forall \ell \in [n], \forall \ell' \in [n], \forall x \in \mathbf{x}_\ell, \forall x' \in \mathbf{x}_{\ell'}, \text{s.t.} \\
& && h(x') \text{ is close to } h(x). \\
& \quad I_{\ell,i} \in \{0,1\}, && \forall \ell \in [n], \forall i \in [\omega_\ell] \\
& \quad I'_{\ell,i} \in \{0,1\}, && \forall \ell \in [n], \forall i \in [\omega_\ell]
\end{aligned}
\tag{1}
$$

We formalize correctness below.

**Proposition 3.8.** *[Optimality] The solution to* (1) *is the optimal solution of Problem 3.7.*

Algorithm 1 summarizes our technique for pruning pre-images from a set NESY training samples. The algorithm works on mini-batches, i.e., it solves (1) for each mini-batch of $\mathcal{D}$.

## 3.3 DISCUSSION

Our formulation in (1) does not consider the strength of the similarity (e.g., the inverse distance) of two instances. We can change the optimization objective to include the similarity of two instances as the weight of an edge. The second point concerns the optimality of the gold proximity graphs for $\mathcal{D}$. Proposition 3.5 states that for each training sample, $\Pi(\mathcal{G}^*_{\mathcal{D}})$ includes the gold pre-image. However, it does not provide an optimality guarantee of the form: There does not exist any other proximity graph $\mathcal{G}'_{\mathcal{D}}$ for $\mathcal{D}$ subject to any encoder $h$, such that the $\ell$-th training sample in $\Pi(\mathcal{G}'_{\mathcal{D}})$ includes the gold pre-image and has fewer pre-images than in $\Pi(\mathcal{G}^*_{\mathcal{D}})$, for some $\ell \in [n]$. This optimality guarantee is not possible unless we make certain assumptions about $\mathcal{D}$ and the edges in $\Pi(\mathcal{G}^*_{\mathcal{D}})$.

The above reveals the third point: Ideally, we should consider all training samples in $\mathcal{D}$ when solving (1). If this is not possible due to scalability restrictions when $\mathcal{D}$ is very large, we should consider a sufficiently large batch size to avoid phenomena in which certain instances have very few or even no other instance of the same class and, hence, there are not enough edges that could potentially filter out pre-images. In our empirical analysis, we saw that reasonably large batch sizes were sufficient to prevent these phenomena. Fourth, as stated in Section 1, our approach can run in a training-free manner or by simultaneously updating the encoder $h$ during training. In all cases, we can apply CLIPPER either on whole $\mathcal{D}$ or on mini-batches, as in Algorithm 1.

The last point concerns the guarantees on preserving the gold pre-images: Proposition 3.5 offers such guarantees; but the formulation of Problem 3.7 does not focus on this aspect. From Proposition 3.5, it follows that offering guarantees on preserving the gold pre-images straightforwardly relates to the

Table 1: Classification accuracy for SUM-$M$.

| Algorithms | $n$=100, MNIST | | $n$=500, MNIST | | $n$=5K, CIFAR-10 | | $n$=10K, CIFAR-10 | |
| | $M = 3$ | $M = 4$ | $M = 3$ | $M = 4$ | $M = 3$ | $M = 4$ | $M = 3$ | $M = 4$ |
|---|---|---|---|---|---|---|---|---|
| SCALLOP | $36.17 \pm 13.28$ | $32.26 \pm 11.48$ | $95.10 \pm 0.28$ | $95.94 \pm 0.00$ | $64.29 \pm 2.93$ | $48.62 \pm 15.74$ | $85.73 \pm 0.26$ | $82.30 \pm 3.47$ |
| + C(GOLD) | $\mathbf{49.36 \pm 8.96}$ | $\mathbf{41.89 \pm 12.11}$ | $95.56 \pm 0.28$ | $\mathbf{96.38 \pm 0.20}$ | $\mathbf{68.94 \pm 1.90}$ | $48.23 \pm 4.47$ | $\mathbf{86.85 \pm 1.41}$ | $82.85 \pm 3.71$ |
| + C(ENC) | $45.25 \pm 12.35$ | $36.93 \pm 11.75$ | $\mathbf{95.95 \pm 0.37}$ | $96.22 \pm 0.20$ | $66.43 \pm 0.67$ | $\mathbf{70.01 \pm 0.70}$ | $86.25$ | $\mathbf{84.60 \pm 0.42}$ |
| DOLPHIN | $35.31 \pm 13.99$ | $\mathbf{32.44 \pm 6.31}$ | $\mathbf{95.16 \pm 0.22}$ | $95.44 \pm 0.34$ | $66.96 \pm 2.85$ | $49.06 \pm 9.75$ | $83.86 \pm 1.14$ | $78.94 \pm 2.96$ |
| + C(GOLD) | $\mathbf{50.54 \pm 8.66}$ | $30.93 \pm 1.13$ | $95.14 \pm 0.50$ | $95.84 \pm 0.28$ | $\mathbf{71.16 \pm 0.60}$ | $38.17 \pm 15.59$ | $\mathbf{85.33 \pm 0.28}$ | $80.63 \pm 1.16$ |
| + C(ENC) | $46.64 \pm 6.89$ | $30.14 \pm 0.65$ | $94.96 \pm 0.30$ | $\mathbf{95.98 \pm 0.38}$ | $66.02 \pm 1.24$ | $\mathbf{67.47 \pm 0.64}$ | $81.50 \pm 0.30$ | $\mathbf{84.08 \pm 0.61}$ |
| ISED | $7.98 \pm 2.82$ | $9.77 \pm 3.12$ | $70.55 \pm 8.39$ | $37.01 \pm 6.08$ | $16.85 \pm 4.10$ | $10.08 \pm 3.17$ | $45.71 \pm 6.76$ | $18.6 \pm 4.31$ |
| + C(GOLD) | $10.28 \pm 3.20$ | $\mathbf{10.51 \pm 3.24}$ | $\mathbf{70.6 \pm 8.40}$ | $60.38 \pm 7.77$ | $\mathbf{33.38 \pm 5.78}$ | $\mathbf{17.06 \pm 4.13}$ | $\mathbf{52.06 \pm 7.21}$ | $\mathbf{29.7 \pm 5.44}$ |
| + C(ENC) | $\mathbf{11.7 \pm 3.42}$ | $9.22 \pm 3.03$ | $70.1 \pm 8.37$ | $\mathbf{70.01 \pm 8.36}$ | $17.43 \pm 4.17$ | $14.26 \pm 3.78$ | $37.06 \pm 6.08$ | $25.91 \pm 5.09$ |
| ABLSIM | $16.51$ | $11.71$ | Running | Running | Running | Running | Running | Running |
| ABLKIT | $16.1$ | Running | Running | Running | Running | Running | Running | Running |

Table 2: Classification accuracy for MAX-$M$.

| Algorithms | $M$=3, $n = 100$ | $M$=4, $n = 100$ |
|---|---|---|
| SCALLOP | $\mathbf{58.19 \pm 3.47}$ | $\mathbf{53.92 \pm 3.28}$ |
| + C(GOLD) | $48.19 \pm 2.86$ | $38.93 \pm 3.13$ |
| + C(ENC) | $43.82 \pm 5.25$ | $35.78 \pm 5.21$ |
| DOLPHIN | $61.86 \pm 2.54$ | $59.70 \pm 6.43$ |
| + C(GOLD) | $\mathbf{65.87 \pm 4.72}$ | $\mathbf{65.30 \pm 4.19}$ |
| + C(ENC) | $63.57 \pm 3.41$ | $61.93 \pm 1.94$ |
| ISED | $9.78 \pm 3.13$ | $\mathbf{9.87 \pm 3.14}$ |
| + C(GOLD) | $8.9 \pm 2.98$ | $8.76 \pm 2.96$ |
| + C(ENC) | $\mathbf{12.63 \pm 3.55}$ | $9.55 \pm 3.09$ |
| ABLSIM | $42.48$ | Running |

Table 3: Classification accuracy for HWF-7.

| Algorithms | $n$=500 | $n$=1k |
|---|---|---|
| SCALLOP | $55.67 \pm 7.93$ | $62.36 \pm 41.83$ |
| + C(GOLD) | $\mathbf{95.01 \pm 0.07}$ | $\mathbf{97.57 \pm 0.06}$ |
| + C(ENC) | $92.46 \pm 0.06$ | $95.95 \pm 0.36$ |
| DOLPHIN | $11.81 \pm 1.94$ | $15.35 \pm 1.11$ |
| + C(GOLD) | $\mathbf{77.52 \pm 5.08}$ | $85.39 \pm 11.97$ |
| + C(ENC) | $66.33 \pm 17.45$ | $\mathbf{89.84 \pm 4.30}$ |
| ISED | $19.33 \pm 4.83$ | $24.44 \pm 8.39$ |
| + C(GOLD) | $\mathbf{28.10 \pm 4.21}$ | $29.42 \pm 4.86$ |
| + C(ENC) | $21.86 \pm 2.63$ | $\mathbf{29.52 \pm 2.77}$ |

"quality" of the edges, that is whether the connected instances are, in fact, of the same class. A way to address this issue would be to associate with each edge $(\ell, x) \to (\ell', x')$ the probability instances $x$ and $x'$ are of the same class. However, this modification is not straightforward due to the correlations between the instances in each training sample. We leave this aspect as a direction for future research.

## 4 EXPERIMENTS

**Benchmarks.** We consider a wide range of benchmarks. The first two, **SUM-$M$** and **MAX-$M$**, are two classic benchmarks in the literature Manhaeve et al. (2021). SUM-$M$ has been used in our running example, while MAX-$M$ considers the maximum instead of the sum of the gold labels. In the above scenarios, the number of pre-images may be particularly large, making the supervision rather weak, e.g., in the MAX-4 scenario, there are $4 \times 9^3$ candidate combinations of labels when the weak label is 9. To assess the effectiveness of our technique under more complex representations, we also consider a variant of those benchmarks, where we associate each digit in $\{0, \dots, 9\}$ with a CIFAR-10 class. The next benchmark is **HWF-$M$** Li et al. (2020). Each training sample consists of (1) a sequence $(x_1, \dots, x_K)$ of digits in $\{0, \dots, 9\}$ and mathematical operators in $\{+, -, *\}$, corresponding to a mathematical expression of length $M$ and (2) the result of the corresponding mathematical expression. The goal is to train a classifier to recognize digits and operators.

Our third benchmark is **VQAR** Huang et al. (2021a). VQAR extends GQA Hudson & Manning (2019) with queries that require multi-hop reasoning using knowledge from CRIC Gao et al. (2019). The benchmark includes the classifiers $name$ and $rel$ that return the type of an object within a given bounding box and the relationship between the objects within a pair of bounding boxes. The objective is to train the above classifiers using samples of the form $(\mathbf{o}, \phi)$, where $\mathbf{o}$ are bounding boxes and $\phi$ is a sentence the bounding boxes abide by. The benchmark includes 500 object types and 229 different relations. We restrict to the top-$k$ most frequent object types and relations for $k = \{50, 100\}$.

Our last benchmark is **MUGEN** Hayes et al. (2022). MUGEN is based on CoinRun Cobbe et al. (2019). Each training sample consists of a sequence of $N$ video frames and a sequence of $K$ actions that describe what the character does. The

Table 4: name (N)/relation (R) classification accuracies for VQAR. i3 and i5 denote the types of sentences in $\mathcal{D}$.

| Algorithms | top-50 N, top-50 R $n = 1000$ | top-100 N, top-50 R $n = 1000$ | top-100 N, top-50 R $n = 5000$ |
|---|---|---|---|
| SCALLOP | 46.58/19.93 | 35.6/$\mathbf{12.62}$ | 37.98/13.94 |
| + C(ENC) | $\mathbf{48.08/22.41}$ | $\mathbf{36.17}$/12.55 | $\mathbf{39.70/14.32}$ |

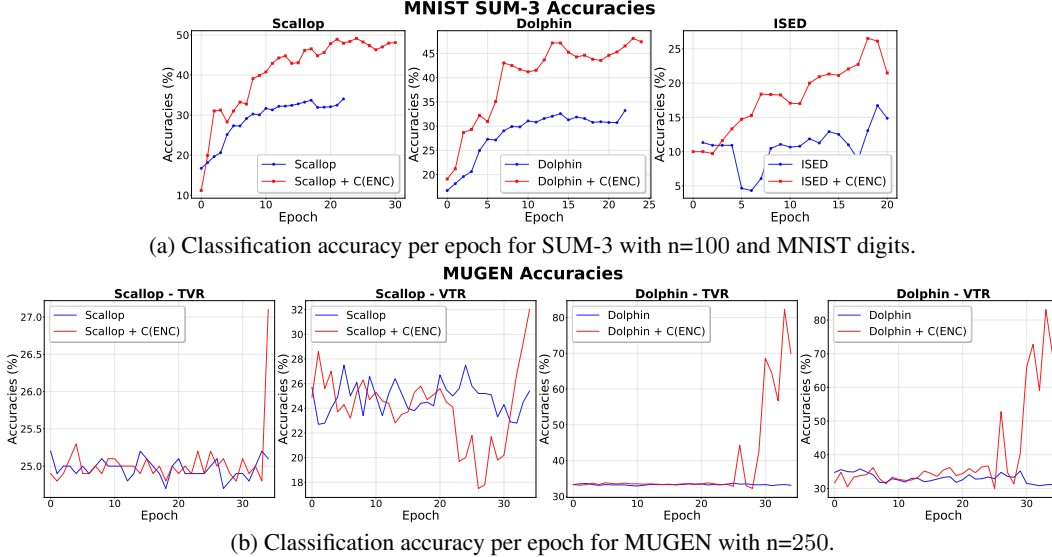

(a) Classification accuracy per epoch for SUM-3 with n=100 and MNIST digits.

(b) Classification accuracy per epoch for MUGEN with n=250.

Figure 1: Classification accuracies per epoch with and without CLIPPER.

objective is to train a classifier to recognize the action in each frame. In general, $K <= N$, i.e., the same action may be taking place in more than one video frame. However, we do not exactly know which action takes place in each frame. We use two tasks to assess the performance of the classifier: video-to-text retrieval (VTR) and text-to-video retrieval (TVR). In VTR, given a video and $M$ sequences of actions, the classifier must choose the sequence of actions most aligned with the video. In TVR, given a sequence of actions and $M$ videos, the classifier must choose the video most aligned with the action sequence. In each task, we measure accuracy by counting the number of times the classifier chose the ground-truth sequence of actions and videos.

**Baselines, Engines, Variants, & Measures.** We consider the state-of-the-art engines SCAL-LOP Huang et al. (2021a), DOLPHIN Naik et al. (2025), and ISED Solko-Breslin et al. (2024). Unlike SCALLOP and DOLPHIN, ISED implements sampling-based NESY learning. We apply CLIPPER, abbreviated as C, using (1) the gold proximity graph for the input training samples and (2) the proximity graph subject to pre-trained encoders. The first setting allows us to assess the potential of our technique independently of the encoder in use. We denote the first setting by C(GOLD) and the second one by C(ENC). In the appendix, we provide details about the encoder used in each benchmark. In MUGEN, we did not have access to the gold labels; we approximated C(GOLD) using a pretrained encoder. We assess the performance of CLIPPER using the classification accuracy of the underlying classifiers. In SUM-$M$, MAX-$M$, and HWF-7, the results are obtained over three runs. In VQAR and MUGEN, each experiment was run once, following Li et al. (2023c). In the first four benchmarks, we run Algorithm 1 in a training-free fashion. In MUGEN, we interleave pruning with the training of the underlying encoder: while in the previous scenarios we had already had access to pre-trained models, this was not the case in MUGEN. We additionally compare with ABLSIM Huang et al. (2021b), a technique that uses similarity-based consistency optimization to prune preimages in Abductive Learning Dai et al. (2019), and ABLKIT Huang et al. (2024b), an efficient Python toolkit for Abductive Learning.

The results of our analysis are shown in Tables 1-5 and Figure 1. The tables show the final classification accuracies after convergence, while the figure shows changes in classification accuracy over training epochs. Additional information is in the appendix.

We see that CLIPPER can significantly increase the accuracy of the baseline model. For example, in SUM-$M$, Table 1, the mean classification accuracy of the baseline SCALLOP model increases from 36.17% to 45.25% when $M = 3$,

Table 5: Classification accuracy for MUGEN.

| Algorithms | VTR | | TVR | |
|---|---|---|---|---|
| | $n$=250 | $n$=500 | $n$=250 | $n$=500 |
| SCALLOP | 25.2 | 26.39 | 27.5 | 31.10 |
| + C(GOLD) | 25.7 | **76** | 28.4 | 76.7 |
| + C(ENC) | **27.1** | 74.9 | **32.00** | **77.7** |
| DOLPHIN | 33.8 | 33.9 | 34.8 | 34.7 |
| + C(GOLD) | 76.1 | 84 | 80.0 | 86 |
| + C(ENC) | **83.7** | **86.6** | **85.0** | **86.9** |

$n = 100$, and MNIST digits are used. For the baseline DOLPHIN and ISED models, it increases from 35.31% to 46.46% and from 8.56% to 8.75%. For HWF-7 in Table 3, the mean accuracy of the baseline SCALLOP model increases from 62.36% to 95.95% when n=1k; for the same scenario, the accuracy for the baseline DOLPHIN model increases from 15.35% to 89.84%. For MUGEN in Table 5, VTR increases for the baseline SCALLOP model from 26.39% to 74.9% when $n = 500$; for DOLPHIN, this increase is from 33.9% to 86.6%. Notice that the baseline accuracy for SCALLOP and the improvements due to CLIPPER that are reported in Table 4 are less than the results in Huang et al. (2021a). This is because of two reasons: (1) Huang et al. (2021a) does not report classification accuracy, but accuracy on the overall output and (2) in Huang et al. (2021a), the samples in $\mathcal{D}$ are easier than those in our analysis. In fact, in our analysis, the training samples are associated with more than 100 pre-images on average.

Out of the 47 scenarios in our empirical results, CLIPPER decreases the accuracy in six scenarios only, and in most cases the decrease is minor. One of these cases are met in ISED. We attribute these minor decreases to the fact that ISED chooses pre-images in a non-deterministic way. Due to this randomness, the gold labels may be discarded during training even if CLIPPER maintains them. Two such cases are met in SCALLOP, under the MAX-$M$ scenario. We attribute this decrease in training instabilities to the way SCALLOP implements the aggregate semiring Li et al. (2023c).

Another observation is that pruning under non-gold proximity graphs leads to results that are on par with those obtained by pruning under the gold proximity graphs. For example, in SUM-3, for $n = 100$, SCALLOP+C(GOLD) improves the mean accuracy by 13% over the baseline; SCALLOP+C(ENC) improves the accuracy by 9%, see Table 1. In ISED, C(ENC) leads to higher improvements than C(GOLD) in most scenarios. As noted above, this is due to the non-deterministic sampling of the pre-images after pruning. For MUGEN, the accuracy of the baseline DOLPHIN model increases from 33.9% to 84% for $n = 500$ under pruning guided by the gold proximity graph. Instead, when a non-gold proximity graph is employed, the accuracy for the same scenario increases from 33.9% to 86.6%. This is because, in MUGEN, we approximated C(GOLD) using a pretrained encoder, i.e., the labels that we use to compute the gold proximity graph are noisy.

The results in Table 5 manifest the robustness of CLIPPER, showing that it performs quite well when there is no access to a pre-trained encoder. Figure 1(b) suggests another interesting phenomenon about CLIPPER: as epochs increase, CLIPPER keeps improving the classification accuracy, as opposed to the baseline. In SCALLOP+C(ENC) and DOLPHIN+C(ENC), we can see drastic increases in TVR and VTR after epochs 30 and 25.

## 5 RELATED WORK

**NESY Engines.** Research on NESY mainly focused on developing efficient NESY losses Xu et al. (2018); Donadello et al. (2017) and sampling-based training techniques Li et al. (2023b); Solko-Breslin et al. (2024); Dai et al. (2019), leaving unexplored the connection between those losses and their ability to disambiguate the gold labels.

**NESY Learning.** Our work was motivated by recent research showing that learning becomes more challenging as the space of possible label combinations increases in size Marconato et al. (2023); Tsamoura et al. (2025). Marconato et al. (2023) proposed different strategies to anticipate the lack of gold labels during training. However, most of these strategies made additional assumptions during training. The only strategy in Marconato et al. (2023) that exploited the representation space was the one employing an autoencoder-based loss during training (Section 5.3). Unlike ours, this strategy requires modifying the classifier's architecture. More importantly, it does not operate in a training-free manner as CLIPPER. Finally, the work in Marconato et al. (2024) that concurrently trains multiple classifiers to improve label disambiguation during training. The latter research is orthogonal to ours.

**Partial Label Learning.** NESY extends *partial label learning* (PLL) Cour et al. (2011). In PLL, each training sample is of the form $(x, Y)$, where $Y$ is a set of mutually exclusive candidate labels for $x$ that includes the gold one. Most relevant to ours is the work of He et al. He et al. (2024) in standard PLL. However, their formulation (1) cannot be extended to support NESY and (2) neither discards the maximum number of pre-images across all training samples, as Proposition 3.8 does – their proposed technique greedily eliminates the candidate labels from $Y$ that do not occur frequently in the top-$k$ neighbors of $x$ in the representation space. Theorem 1 in He et al. (2024) computes the probability of discarding the gold labels as a function of the total number of discarded labels.

However, their analysis cannot be straightforwardly extended to analyze our technique due to the correlations among instances in each training sample – which are not supported by their technique.

## 6 CONCLUSIONS

We introduced a technique to reduce the space of candidate label combinations in NeSy by exploiting the proximity of instances in the representation space that is supported by a new problem formulation and an optimal solution. An option would be to use pretrained LLMs to infer the gold labels directly Stein et al. (2025). Beyond being cost- and resource-demanding, this approach can be seen as a special case of our approach that retains only one pre-image: the one that best aligns with the LLM's predictions. Our formulation is a non-trivial extension to this setting, where we do not keep a single pre-image but multiple ones that abide by the background theory. Future research includes extending the theoretical analysis in He et al. (2024) to our setting.

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

## A  NOTATION

Table 6: The notation in the preliminaries and the proposed algorithm.

| Supervised learning notation | |
|---|---|
| $[n] := \{1, \ldots, n\}$ | Set notation |
| $\mathcal{X}, \mathcal{Y}$ | Input instance space and label space |
| $x, y$ | Elements from $\mathcal{X}$ and $\mathcal{Y}$ |
| $\Delta_c$ | Space of probability distributions over $\mathcal{Y}$ |
| $f : \mathcal{X} \mapsto \Delta_c$ | Scoring function |
| $f^j(x)$ | Score of $f$ upon $x$ for class $j \in \mathcal{Y}$ |
| **NSL notation** | |
| $\mathcal{D}$ | Set of NESY training samples |
| $n$ | Number of samples in $\mathcal{D}$ |
| $\ell, \ell'$ | Indices over $[n]$ |
| $\mathbf{x}_\ell$ | The vector of instances in the $l$-th NESY sample in $\mathcal{D}$ |
| $\sigma_{\ell,i}$ | The $i$-th pre-image of the $\ell$-th NESY sample in $\mathcal{D}$ |
| $\omega_\ell$ | Number of pre-images in the $\ell$-th NESY sample in $\mathcal{D}$ |
| **Notation in Section 3** | |
| $h$ | Encoder from $\mathcal{X}$ to $\mathbb{R}^m$ |
| $\mathcal{G}^h_\mathcal{D}$ | Proximity graph for $\mathcal{D}$ subject to $h$ |
| $E_{\ell,\ell',x,x'}$ | Binary variable becoming 1 if $(\ell, x) \to (\ell', x')$ is in $\mathcal{G}^h_\mathcal{D}$ |
| $I_{\ell,i}, I'_{\ell,i}$ | Binary variables corresponding to pre-image $\sigma_{\ell,i}$ in $\mathcal{D}$ |

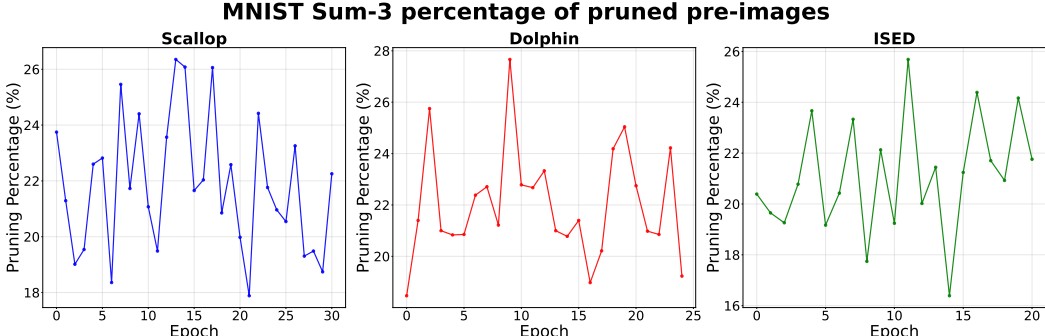

Figure 2: Percentage of pre-images pruned by CLIPPER across training epochs for MNIST SUM-3 with $n = 100$.

## B  DETAILS ON SECTION 3

**Proposition 3.5.** *For each $\ell \in [n]$, the $\ell$-th training sample in $\Pi(\mathcal{G}_{\mathcal{D}}^*)$ includes the gold pre-image.*

*Proof.* The proof proceeds by means of contradiction. Suppose that in the $\ell$-th training sample, the gold pre-image $sigma_\ell^*$ is not retained in $\Pi(\mathcal{G}_{\mathcal{D}}^*)$. According to Definition 3.4, the above implies that $sigma_\ell^*$ is inconsistent in $\mathcal{G}_{\mathcal{D}}^*$. According to Definition 3.4, the latter consequently means that $\mathcal{G}_{\mathcal{D}}^*$ includes an edge $(\ell, x) \to (\ell', x')$ $\sigma_\ell^*$ is inconsistent with, i.e., there does not exist any pre-image $\sigma_{\ell',i'}$, such that $\sigma_\ell^*(x) = \sigma_{\ell',i'}(x')$ holds. Let $c$ be the gold label of $x$. Since $\mathcal{G}_{\mathcal{D}}^*$ is a gold proximity graph for $\mathcal{D}$, it follows that $x$ and $x'$ belong to the same class $c$. Furthermore, since $\sigma_\ell^*$ is inconsistent with $(\ell, x) \to (\ell', x')$, this means that no pre-image in the $\ell'$-th training sample maps $x'$ to $c$. However, by definition, each NESY training sample includes the gold pre-image, i.e., the image that maps each instance to each ground truth class. The above leads to a contradiction, completing the proof of Proposition 3.5. $\square$

**Proposition 3.8.** *[Optimality]  The solution to* (1) *is the optimal solution of Problem 3.7.*

*Proof.* The proof proceeds by construction. Recall that the variable $I'_{\ell,i}$ corresponds to the $i$-th pre-image of the $\ell$-th training sample $\sigma_{\ell,i}$, for $\ell \in [n]$ and $i \in [\omega_\ell]$ and becomes one if $\sigma_{\ell,i}$ is *not* in the pruning of $\mathcal{D}$ subject to the resulting proximity graph, see Section 3.2. Due to the above, the optimization objective $\max \sum_{\ell \in [n], i \in [\omega_\ell]} I'_{\ell,i}$ in (1) aligns with the objective (3) in Problem 3.7.

Now, let us move to objective (2) from Problem 3.7, that is the pruning of $\mathcal{D}$ subject to the resulting proximity includes all globally consistent pre-images. This objective is satisfied due to the third constraint in (1).

Finally, let us move to objective (1) from Problem 3.7, that is the pruning of $\mathcal{D}$ subject to the resulting proximity is sound, that is, each at least one pre-image is preserved for each sample, see Definition 3.4. From Section 3.2, we know that the variable $E_{\ell,\ell',x,x'}$, for each $\ell, \ell' \in [n]$, $x \in \mathbf{x}_\ell$, and $x' \in \mathbf{x}'_\ell$, (a) denotes that $h(x')$ is close to $h(x)$ and (b) becomes one if the resulting proximity graph includes the corresponding edge and zero otherwise. The fourth constraint in (1), that is, $1 - E_{\ell,\ell',x,x'} + 1 - I_{\ell,i} = 1$, enforces that each pre-image $\sigma_{\ell,i}$ that is inconsistent with the edge $(\ell, x) \to (\ell', x')$ (and this edge is included in the resulting proximity graph) will not be included in the pruning of $\mathcal{D}$ subject to the resulting graph. Notice that whenever the variable $E_{\ell,\ell',x,x'}$ becomes one, the variable $I_{\ell,i}$ becomes zero. The above, along with the facts that (i) each training sample in the resulting pruning of $\mathcal{D}$ is associated with at least one pre-image – enforced by the constraint $\sum_{i=1}^{[\omega_\ell]} I_{\ell,i} \geq 1$, for each $\ell \in [n]$ in (1) – and (ii) each pre-image is either included in the resulting pruning or not – enforced by the constraint $I_{\ell,i} + I'_{\ell,i} = 1$, for each $\ell \in [n]$ and each $i \in [\omega_\ell]$ in (1) – ensure that the LP in (1) satisfies objective (1) from Problem 3.7, completing the proof of Proposition 3.8. $\square$

## C    FURTHER DETAILS ON THE EXPERIMENTS

**Benchmarks.** In **SUM**-$M$ and **MAX**-$M$ training samples are created by drawing $M$ MNIST digits or CIFAR-10 images in an i.i.d. fashion and associating with them the sum or maximum of their corresponding gold labels. Regarding **VQAR** Huang et al. (2021a), the original benchmark includes a large number of queries that reduce NESY training to supervised one. To make training more challenging, we consider training samples associated with a large number of pre-images, averaging on more than 100 per training sample. To control the difficulty of training, we consider training samples whose logical formulae are of the form $name(superclass, o_1) \land rel(r, o_1, O_2)$, where $superclass$ is the most generic class object $o_1$ can belong to according to the CRIC ontology, e.g., a toy is an object, and $a$ is the attribute object $O_2$ is associated with. The above formulae are slightly different than the ones described in Section 2, as they include free variables: $o_1$ is a given object, while $O_2$ is a free one that can be found to any object within a given image. Our analysis applies without loss of generality to those settings due to the flexibility abduction gives us.

**Engines.** Like DeepProbLog, SCALLOP relies on training using semantic loss Xu et al. (2018). However, it offers a scalable implementation of it. Research showed that SCALLOP outperforms DeepProbLog, ABL Dai et al. (2019), NeurASP Yang et al. (2020) and the engine proposed in Li et al. (2023b) across a variety of tasks Wang et al. (2023); Li et al. (2023c). In addition, SCALLOP has state-of-the-art performance on MUGEN and VQAR, outperforming SDSC Hayes et al. (2022) in MUGEN, and NMNs Andreas et al. (2016) and LXMERT Tan & Bansal (2019) in VQAR. DOLPHIN offers NESY training using losses based on fuzzy logic and has reported higher accuracy than SCALLOP on a variety of benchmarks.

**Encoders.** We use a pretrained ResNet-18 convolutional neural network pretrained on ImageNet-1k for SUM-$M$, MAX-$M$, CIFAR-10, and HWF. We show the amount of proofs pruned by CLIPPER with the ResNet-18 encoder in Figure 2. In VQAR, the object bounding boxes and features are obtained by passing the images through pre-trained fixed-weight Mask RCNN and ResNet models. For MUGEN, as discussed in the experiments section, we do not use an external encoder, but rather use the model to encode the representations while it is being trained. To approximate the gold labels, we trained the same model on the full MUGEN dataset until convergence to produce an oracle model $M_O$, which we used to approximate the gold labels.

**Computational environment.** All experiments, except MUGEN, were performed on machines with two 20-core Intel Xeon Gold 6248 CPUs, four NVIDIA GeForce RTX 2080 Ti (11 GB) GPUs, and 768 GB RAM. MUGEN, as it required a larger GPU, was trained on an NVIDIA A100 40GB GPU.

**Additional implementation details.** Across all experiments, we deal with directed proximity graphs and assume that there exists a directed edge $(\ell, x) \to (\ell', x')$ only if $h(x')$ is in the top-$k$ neighborhood of $h(x)$, for $x, x' \in \mathcal{X}$, for $k = 1$. We use a batch size of 64 for SUM-$M$ and MAX-$M$, with a learning rate of $1e - 3$. For HWF, we used a batch size of 4 with a learning rate of $1e - 4$. For VQAR, we used a batch size of 512 with a learning rate of $1e - 4$. For MUGEN, we used a batch size of 3 for Dolphin and 4 for Scallop, with a learning rate of $1e - 4$. Across all experiments, we used AdamW as the optimizer. To compute proximity in the latent space, we used the FAISS open source library Johnson et al. (2021).

**Software packages.** Our source code was implemented in Python 3.10. We used the following python libraries: `scallopy`[3], `highspy`[4], `or-tools`[5], `PySDD`[6], `PyTorch` and `PyTorch vision`.

**Deep networks.** For MAX-$M$ and SUM-$M$, we used the MNIST CNN used in (Huang et al., 2021a). For HWF-7, we used the CNN used in (Li et al., 2023c). For VQAR and MUGEN, we used the same deep networks with (Huang et al., 2021a).

Figure 2 shows the percentage of pre-images that are pruned at each epoch for MNIST SUM-3 with $n = 100$. We can see that CLIPPER is quite effective in discarding pre-images, discarding more than 25% on average across all engines.

---

[3] https://github.com/scallop-lang/scallop (MIT license).
[4] https://pypi.org/project/highspy/ (MIT license).
[5] https://developers.google.com/optimization/ (Apache-2.0 license).
[6] https://pypi.org/project/PySDD/ (Apache-2.0 license).

Table 7: Ablations for MNIST Sum-3, $n = 100$ for different encoders and batch sizes. In the rows "+ C(RESNET18)" and "+ C(RESNET50)", the encoder is pretrained and frozen. In the row "+ C(MNISTNET)", the encoder is MNISTNet, the underlying CNN being trained, and is randomly initialized and trainable.

| Batch Size | Algorithms | Classification Accuracy | % Retained Preimages | % Preimages with Ground Truth | Time to Prune (s) | Time to Solve ILP (s) |
|---|---|---|---|---|---|---|
| 64 | Dolphin | 31.6 | NA | NA | NA | NA |
| | + C(RESNET18) | 47.09 | 77.99 | 91.56 | 0.49 | 0.03 |
| | + C(RESNET50) | 41.89 | 75.7 | 88.38 | 0.51 | 0.03 |
| | + C(MNISTNET) | 41.51 | 78.12 | 94.81 | 0.53 | 0.03 |
| 128 | Dolphin | 32.86 | NA | NA | NA | NA |
| | + C(RESNET18) | 49.24 | 80.17 | 92 | 1.88 | 0.06 |
| | + C(RESNET50) | 42.56 | 74.21 | 90 | 1.5 | 0.07 |
| | + C(MNISTNET) | 36.84 | 77.85 | 96.21 | 1.39 | 0.07 |

Table 8: Ablations for MNIST Sum-4, $n = 100$ for different encoders. In the rows "+ C(RESNET18)" and "+ C(RESNET50)", the encoder is pretrained and frozen. In the row "+ C(MNISTNET)", the encoder is MNISTNet, the underlying CNN being trained, and is randomly initialized and trainable.

| Batch Size | Algorithms | Classification Accuracy | % Retained Preimages | % Preimages with Ground Truth | Time to Prune (s) | Time to Solve ILP (s) |
|---|---|---|---|---|---|---|
| 64 | Dolphin | 31.43 | NA | NA | NA | NA |
| | + C(RESNET18) | 31.28 | 87.5 | 96.19 | 3.39 | 0.39 |
| | + C(RESNET50) | 28.69 | 86.46 | 92.9 | 3.42 | 0.42 |
| | + C(MNISTNET) | 35.19 | 85.88 | 98.58 | 3.34 | 0.39 |

# D ABLATIONS

We run ablations on the complexity of encoders used for extracting latent representations used to prune preimages. We also measure the amount of preimages pruned in each case, the amount of preimages where the ground truth was retained, and the time taken for pruning. We report the results of ablations on MNIST Sum-3 and Sum-4 with 100 training samples in Tables 7 and 8.

## D.1 PERFORMANCE ON FULL DATASETS.

We are running our experiments on the full dataset versions of each benchmark. Table **??** shows the results of MNIST Sum-3 on the full dataset. We also run our MUGEN experiment on the full dataset with 5000 training samples. We show preliminary accuracy curves in Figure 3.

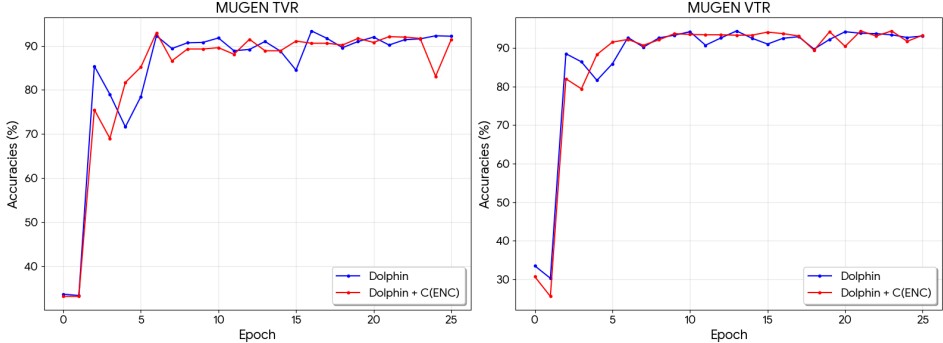

Figure 3: Training curves for Dolphin with and without the encoder for MUGEN with 5K training samples.

Table 9: Results of CLIPPER for the full versions over our benchmarks.

| Algorithms | MNIST Sum-3 |
|---|---|
| DOLPHIN | 99.11 |
| + C(RESNET18) | 99.13 |
| ABLSIM | 98.9 |
| ABLKIT | 98.2 |