# OpenReview forum: "On Improving Neurosymbolic Learning by Exploiting the Representation Space"
_ICLR.cc/2026/Conference — Submitted to ICLR 2026_

### Official Review · Reviewer_jDFH · 2025-10-30

**Soundness:** 3
**Presentation:** 3
**Contribution:** 3
**Rating:** 6
**Confidence:** 3

**Summary:**

This paper addresses a key challenge in neurosymbolic learning (NESY): the exponential growth of possible label combinations (pre-images) that satisfy the logical constraints during training, which weakens supervision. The authors propose a novel technique, CLIPPER, which prunes this space of candidate pre-images by leveraging the intuition that instances with similar latent representations should share the same label. The core of the method involves constructing a proximity graph based on instance representations and formulating the pruning problem as an Integer Linear Program to discard inconsistent pre-images while ensuring no training sample loses all its candidates. The approach is framework-agnostic and can be applied pre-training or during training. Experiments on three neurosymbolic engines across multiple benchmarks show substantial accuracy improvements.

**Strengths:**

1. Novelty is good. The work is the first to systematically exploit the representation space to prune pre-images in a general neurosymbolic learning setting. While similar intuitions exist in partial label learning (PLL), their application under complex logical constraints is novel and non-trivial. The technical approach is well-motivated and formally grounded. The formulation of the pruning problem as an ILP is sound and provides a clear, optimizable objective.

2. The paper is well-written. The running example effectively illustrates the core problem and the proposed solution. The definitions (proximity graph, consistency, and pruning) are clear and build logically towards the problem statement and the proposed ILP formulation.

3. Experiments. The reported performance gains are impressive and demonstrate the potential of this approach to enhance the scalability and accuracy of neurosymbolic systems.

**Weaknesses:**

1. Computational Complexity. The paper formulates the pruning as an ILP, which is NP-hard in general. For large-scale datasets with many training samples and pre-images, solving this ILP could become computationally prohibitive.

2. Dependence on Encoder Quality. The effectiveness of CLIPPER is inherently tied to the quality of the encoder h. Suppose the encoder produces poor representations (e.g., maps instances of different classes close together). In that case, the proximity graph will be noisy, leading to the pruning of correct pre-images or insufficient pruning.

**Questions:**

As stated in Line 315, "Fourth, as stated in Section 1, our approach can run in a training-free manner or by simultaneously updating the encoder during training." Could you please further elaborate on it?

---

> ### Author Response · Authors · 2025-11-21
>
> ## On computational complexity
>
> Please see our responses to all the reviewers, Section “ILP Overhead” for a detailed discussion and additional results and ablations.
>
> Notice that the number of samples does not impact the runtime of our ILP formulation. Instead, the number of classes and the batch size have an impact on it. For ablations using different batch sizes, please see the new results in Section “ILP Overhead”. Below, we report the number of classes and the maximum number of pre-images per scenario:
> - SUM-3, 10 classes, 54 pre-images on average
> - SUM-4, 10 classes, 440 pre-images on average
> - MAX-3, 10 classes, 171 pre-images on average
> - HWF-7, 13 classes, 200 pre-images on average
> - VQAR, 100 classes, 223.72 pre-images on average
> - MUGEN, 8 classes, 44.48 pre-images on average
>
> ## On the choice of encoder
>
> CLIPPER leads to substantial accuracy improvements in all the following scenarios:
> - Using different pre-trained and frozen encoders: ResNest18, ResNet50, and the standard MNIST classifiers CNN, Section “Robustness of CLIPPER: w.r.t. encoders”.
> - Starting with a randomly initialized encoder that is being updated during NeSy learning, Section “Robustness Of CLIPPER: w.r.t. encoders” in our replies to all reviewers, and Table 5 in the main body of our paper.
> - Using different ways to define proximity in the latent representations, see Section “Robustness Of CLIPPER: w.r.t. different notions of proximity” in our replies to all the reviewers.
>
> ## Explaining training-free manner vs updating the encoder at train time
>
> Thank you for your comment. Training-free manner refers to using a pre-trained and frozen encoder to decide proximity of two different instances and, hence, to form the ILP in Eqn 1. During training, we do not update the weights of the encoder.
>
> By “simultaneously updating the encoder during training”, we refer to the case where we also update the weights of the encoder during training. Of course, to support this scenario, the outputs of the encoder must be connected to the output classification layer. Notice also that:
> - As discussed in lines 409 –412, in the first four benchmarks, we ran Algorithm 1 in a training-free fashion, i.e., we use a *pretrained and frozen* encoder to discard pre-images.
> - In MUGEN (the experiments where we have performance gains of 50\%, Table 5), we start with a *randomly initialized* encoder. During training, we use the encoder to prune pre-images, backpropagating through it at the same time.

---

### Official Review · Reviewer_a97x · 2025-10-30

**Soundness:** 1
**Presentation:** 1
**Contribution:** 3
**Rating:** 2
**Confidence:** 3

**Summary:**

This paper addresses the challenge of neurosymbolic learning (NeSy) in weakly supervised settings, specifically where the vast number of valid label combinations (pre-images) satisfying a logical formula makes learning prohibitively difficult. The authors propose a novel technique, CLIPPER, to prune this exponentially large space of pre-images by exploiting the latent representation space. The core intuition is that instances with similar latent representations should share the same gold label, and this consistency constraint is used to prune the candidate set. The authors claim that this method significantly improves accuracy, particularly in few-shot settings, when applied to state-of-the-art neurosymbolic engines like Scallop, Dolphin, and ISED.

**Strengths:**

The paper tackles a significant and practical challenge in NeSy. The ambiguity arising from a combinatorial explosion of valid pre-images in weakly-supervised settings is a known bottleneck for learning.

**Weaknesses:**

Weaknesses

My main concern with this paper lies in the experimental evaluation, which feels highly specialized and potentially misleading, casting doubt on the general applicability and true performance gain of the proposed method.

Suspiciously Low Baseline Performance: The reported baseline accuracies for SOTA models like Dolphin and Scallop are extremely low (e.g., ~30-35% on SUM-M and MUGEN). This is in stark contrast to the performance reported in their original papers (e.g., the Dolphin paper), where these models achieve >95% accuracy on the same benchmarks.

Artificial Experimental Setting: The reason for this discrepancy seems to be twofold, and both points suggest the experimental setting was intentionally designed to make the baselines fail:

Extreme Few-Shot Setting: The authors appear to have used an extremely small number of data points (e.g., $n=100$ for SUM-M, $n=250$ for MUGEN). While this is a valid "few-shot" setup, it is not the standard way these benchmarks are evaluated.

"Hardest" Samples: The authors also state they use samples associated with "more than 100 pre-images on average," creating a "worst-case" scenario of ambiguity.

While CLIPPER shows gains in this specific, harsh environment, this evaluation fails to demonstrate if the method provides any benefit in the standard, full-dataset setting where baselines are known to perform well.

Questionable Experimental Graphs (Figure 1): The graphs in Figure 1 are highly concerning:

The baselines (solid lines) appear to just "not converged," This is because
The proposed method, C(ENC) (dotted lines), shows a sudden, dramatic spike in accuracy (e.g., on MUGEN) immediately before the training is stopped (e.g., after epoch 25). How do you guarentee this rise of performance is maintained, and baseline performance is maintained low after long training?

This is highly suspicious. The MNIST SUM-3 graph (Fig 1a) shows the C(ENC) method's accuracy peaking and then dropping, suggesting instability or overfitting. Stopping the MUGEN experiment right at the peak, and reporting "best acc," hides whether this performance is stable or if it would have collapsed with further training.

The Core Doubt (Lack of Generalizability): The paper's entire premise rests on showing gains in a few-shot setting where baselines fail. However, my strongest suspicion is that if this method were applied to the standard, large-scale benchmark (as used in the Dolphin paper), the performance gain would be negligible or non-existent, as the ambiguity problem is already solved by the large data. Furthermore, the ILP-based pruning likely adds significant computational overhead, making the method slower than the baselines in a fair "10-hour fix" comparison like the one used in the Dolphin paper.

**Questions:**

1. My strongest suspicion is that this method provides no significant gain in a standard setting. The authors must provide a direct comparison against Dolphin using the exact same standard, full-dataset benchmark setting from the Dolphin paper (e.g., MUGEN 5K, HWF-15). This experiment must:
a.  Use the same 10-hour fixed time limit.
b.  Report the final accuracy at 10 hours (not the "best" accuracy).
c.  Report the additional computational overhead (in time per epoch) caused by the ILP-pruning step.

2. Extending Existing Experiments: To address the highly suspicious graphs in Figure 1, the authors must, at a minimum, re-run their existing few-shot experiments (e.g., MUGEN $n=250$, SUM-M $n=100$) for a full 10-hour time limit.
a.  They must provide graphs showing the full 10-hour training run for both the baselines and their method.
b.  They must report the final epoch's accuracy at 10 hours, not the "best" accuracy. This is crucial to verify if the baselines eventually converge and if the C(ENC) spike is stable or merely a volatile peak.

---

> ### Author Response · Authors · 2025-11-21
>
> ## About gains while training with the full dataset
>
> First, we would like to stress that for MUGEN, SUM-3, or other benchmarks we also use in our paper, both Scallop and Dolphin are quite effective when using the *whole* dataset, i.e., they converge after a fixed number of epochs. We are not expecting CLIPPER to bring major performance improvements in such cases, e.g., as in SUM-2 where the baseline Dolphin model reaches 98\% accuracy. However, as our empirical results show, CLIPPER is quite effective in challenging scenarios where convergence is not guaranteed, e.g., limited data scenarios.
>
> Regarding your request, we are currently running the experiment on MUGEN for $n$=5K samples. We have added the accuracy curves to the revised paper. We just got results for the first 8 epochs (see Figure 3 in Appendix D), and we will keep updating the plots as more epochs are completed. We can see that both techniques perform comparably and reach 90% accuracy for both VTR and TVR.
>
> ## On running experiments for a full 10 hours
>
> All the results on MUGEN that are reported in our paper (Table 5) for $n$=250 took almost 10 hours (35 epochs were completed). The *final* accuracy for Dolphin is: for TVR 33.80\%, for VTR 34.8\%. The *final* accuracy for Dolphin + CLIPPER is: for TVR 82.4\%, for VTR 77.9%.
>
> We will also run the other benchmarks for a full 10 hours as suggested by the reviewer.

---

### Official Review · Reviewer_xgWW · 2025-11-03

**Soundness:** 1
**Presentation:** 2
**Contribution:** 1
**Rating:** 2
**Confidence:** 5

**Summary:**

This paper proposes CLIPPER, a method to improve the efficiency of neuro-symbolic learning by pruning redundant label combinations using representation-space similarity. In typical NeSy systems, each instance has multiple logically consistent label combinations, leading to an exponentially large search space. CLIPPER constructs a proximity graph over latent embeddings and formulates pruning as an integer linear programming (ILP) problem that removes inconsistent candidates while ensuring each instance retains at least one valid label. Integrated with systems such as Scallop, Dolphin, and ISED, CLIPPER achieves notable gains across several benchmarks.

**Strengths:**

The proposed approach is technically solid. The integration of representation-space information into neuro-symbolic learning is handled in a careful and systematic way. Experimental results are convincing, showing consistent improvements across multiple benchmarks and engines.

**Weaknesses:**

## Substantial conceptual overlap with NeurIPS 2021 ABLSim; the claimed "first" contribution does not hold.

The paper's stated motivation and contribution closely mirror those of "Fast Abductive Learning by Similarity-based Consistency Optimization" [Huang et al., NeurIPS 2021], sharing both the same problem formulation and the same central idea. In both works, the key challenge lies in the exponentially growing space of candidate label combinations, which makes learning in neurosymbolic systems computationally expensive and difficult to scale.

Building directly upon this context, the authors claim novelty by stating:
> "We propose the first technique that prunes this space by exploiting the intuition that instances with similar latent representations are likely to share the same label."

However, this intuition—the use of latent representation similarity to reduce the candidate label space—was precisely the foundational principle of ABLSim. That earlier work already pruned the space of abduced label combinations based on the idea that samples close in representation space should share the same label, introducing a similarity-based mechanism to guide the pruning process and remove inconsistent label configurations. The present paper employs the same conceptual mechanism, reformulated through an ILP-based global pruning approach rather than a similarity-scoring heuristic or beam search. While the implementation differs in its optimization formalism, the underlying conceptual contribution is identical: both methods exploit latent-space similarity to constrain combinatorial search in neurosymbolic learning.

(For context, the paper studies neural classifiers in a neurosymbolic setting where hidden labels must satisfy a logical formula—precisely the Abductive Learning (ABL) setup [Dai et al., NeurIPS 2019], where a neural perception module predicts latent symbolic labels that must conform to a knowledge base. The authors highlight that the space of label combinations satisfying the formula grows exponentially, making training inefficient. This issue has long been recognized as the main computational bottleneck in NeSy learning (also, ABL), and ABLSim was specifically designed to address it by leveraging representation-level similarity to restrict the candidate label space. In other words, ABLSim already tackled the same combinatorial explosion that this paper presents as its core motivation.)

Given this lineage, **the contribution claim (in lines 60–69) is unsupported**. The conceptual contribution of using latent-space similarity to guide or prune candidate label combinations in neurosymbolic learning was already introduced, formalized, and empirically validated in ABLSim. **While this paper presents a different implementation (an ILP-based formulation and integration with additional NeSy engines), these are implementation-level refinements rather than a genuine conceptual innovation.** The novelty statement should therefore be revised to acknowledge ABLSim’s precedence and to clarify that this work extends an existing idea through a new optimization formulation, rather than being the first to introduce it.

---

Other points:

1. Since the pruning mechanism relies on embedding-space proximity, its effectiveness is strongly tied to how well the encoder captures semantic similarity. However, the paper does not analyze or control for this dependency, with no experiments comparing different encoders or examining performance sensitivity to embedding quality, making it unclear whether the reported improvements stem from the pruning mechanism itself or from stronger underlying representations. Consequently, the empirical results may not substantiate the contribution's key premise that closeness in latent representations reliably implies label equality.

2. *(minor)* The theoretical analysis in this paper, which formulates neuro-symbolic learning as a partial-label learning problem, is not new. Beyond the works mentioned in lines 72–74, approaches applying PLL to NeSy analysis have been presented before. See examples in Section 3.4 of He et al., ICML 2024.

3. *(presentation)* Most of the citation commands (e.g., \citep, \citet and \cite) are not properly used. Most citations should appear in parenthetical form, for example, line 127 should read: *However, in one of the benchmarks that we consider in our experiments, namely VQAR (Huang et al.
2021), K is commonsense knowledge from CRIC (Gao et al. 2019).* Only a few instances should use the textual form, for example, on line 135: *More details on abduction are in Tsamoura et al. (2021).*

---

Huang et al., Fast Abductive Learning by Similarity-based Consistency Optimization. NeurIPS 2021.

Dai et al., Bridging Machine Learning and Logical Reasoning by Abductive Learning. NeurIPS 2019.

He et al., Ambiguity-Aware Abductive Learning. ICML 2024.

**Questions:**

The method involves potentially expensive steps (e.g., proximity graph construction, ILP solving), yet the paper lacks a discussion on runtime or scalability. How efficient is the overall training compared to the baselines, and can the proposed method scale to large datasets?

Other questions see weaknesses.

**Details Of Ethics Concerns:**

The paper shows substantial motivational and conceptual overlap with prior work (NeurIPS 2021 ABLSim), while claiming novelty for essentially the same idea without proper acknowledgment.

---

> ### Author Response · Authors · 2025-11-21
>
> ## About being flagged for plagiarism
>
> The reviewer can see our reply to all reviewers, Section “About Plagiarism”.
>
> ## About the empirical results
>
> The reviewer can see our responses to all reviewers for our preliminary ablations with respect to the encoder (Table 1 in Section “ILP Overhead”).

---

> > ### Comment · Reviewer_xgWW · 2025-11-27
> >
> > > ABLSim is an adhoc technique for NeSy learning. Unlike ABLSim, CLIPPER can be plugged on top of several frameworks, such as Scallop, Dolphin and ISED, leading to substantially higher classification accuracy over the baselines.
> >
> > The similarity-based consistency measure (Eq. 2) in ABLSim is a generic mechanism applicable to any setting where logically constrained label combinations generate a large candidate space. Nothing in its formulation restricts it to a particular architecture or system. I do not see sufficient grounds to characterize it as an "ad-hoc" technique.
> >
> > > ABLSim’s consistency measure (Eq. 12 in Huang et al., NeurIPS 2021) is different from our notion of consistency, see Def 3.2 in our paper.
> >
> > and
> > > Our problem formulation, Problem 3.7 in our paper, and the solution to our problem, Eqn 1 in our paper, is very different from ABLSim’s objective, Eq. 12 in Huang et al., NeurIPS 2021.
> >
> > I am not accusing the authors of textual plagiarism. Based directly on the description in lines 60–64 (the "first contribution" claim) and lines 186–189 (the role of consistency), despite differences in mathematical formalization, both works operationalize the same underlying idea—leveraging latent-space similarity to prune or constrain the space of candidate label combinations under logical constraints. The modeling choices differ, but the conceptual starting point is the same.
> >
> > > We formalize under which cases the gold pre-images can be maintained, ... ABLSim gives NO such guarantees.
> >
> > This is correct, but I share Reviewer yU2T’s concerns about this guarantees claimed for CLIPPER.
> > 1. The guarantees, as you mentioned, "rely on conditions", for example, access to the gold proximity graph, undirected connectivity, and the absence of cross-class edges, whereas it seems impossible by your statement to confirm whether hold in the actual learning setting ("the gold proximity graph is unknown"). These conditions for the guarantee to stant are neither **verifiable** nor **evaluated** in the experiments, and the guarantees cannot explain or justify the improvements of your proposed methods.
> > 2. Same surprise/confusion with reviewer yU2T, even the "gold" labels experiments show undesiable results, this further raises the question of how informative these "gold pre-images can be maintained" guarantees are for the practical behavior of the method? (you responded that these “gold” labels are approximate yet presumably accurate, then I feel confused what this metric is intended to demonstrate.)
> >
> > > As previous research has shown, see Tsamoura et al. (2021) and Wang et al. (2023), ABL is prone to local minima leading to very low classification accuracy compared to other baselines such as DeepProbLog and Scallop. The experimental results of Tsamoura et al. (2021) and Wang et al. (2023) offer a detailed discussion.
> >
> > I also conducted an additional check. According to the newest implementation on their public GitHub account (https://github.com/AbductiveLearning), recent implementations (released on 2024) show competitive advantages over systems like DeepProbLog. Since the paper cites ABL in the **experimental section** and attempts to compare with ABL results, it is unclear why these developments were not taken into account.
> >
> > That said, in spite of the empirical performance, the limitations of earlier ABL implementations do not justify overlooking conceptual predecessors, and revisiting the same problem with the same conceptual starting point does not become a new contribution. using latent-space similarity to guide or prune candidate label combinations in neurosymbolic learning was already introduced, formalized, and empirically validated in the four-year-old method ABLSim. While CLIPPER provides a different implementation (via ILP formulation and integration with multiple NeSy engines), these are implementation-level refinements rather than a genuine conceptual innovation.

---

> > > ### Author Response · Authors · 2025-12-03
> > >
> > > We cited ABLSim in the revised version of our paper and started conducting experiments having it as a baseline. The new results are added in Tables 1 and 2 or our paper. See also Table 3 in our replies to all the reviewers. **In contrast to ABLSim, which decreases accuracy relative to the baseline, our work can substantially improve the baseline's accuracy.**
> > >
> > > Upon the reviewer’s suggestion, we ran the MNIST Sum-3 experiment on the full dataset using **ABLKit (considering the new GitHub repository of ABL)**. ABLKit brought no improvement over Dolphin, Dolphin+CLIPPER(RESNET-18), or even ABLSim. The results on ABLKit are added to Table 1 in our paper. For MNIST Sum-3 and $n=100$, ABLSim and ABLKit performed comparably, at around 16.5\% and 16.1\%, respectively, which is much worse than CLIPPER’s performance at around 45% and 46% when combined with Scallop and Dolphin, respectively. Regardless, we cite both ABLSim and ABLKit.
> > >
> > > As we mentioned in the response to Reviewer yU2T, the gold labels are approximate **only for MUGEN where the gold labels do not exist**. This was also explained in the paper (lines 448–449). We use the true gold labels to prune under the gold proximity graph in all other cases. Our empirical results (both in our paper and the new Tables 1, 2, and 3 in our replies to all the reviewers) show that CLIPPER brings substantial improvements even when pruning without using the gold proximity graphs, i.e., without having access to the gold labels.
> > >
> > > We disagree with the reviewer’s claim of “conceptual plagiarism”. We already cited ABLSim in the revised version of our paper, and have also discussed it in our related work. Using a fundamentally different formulation to solve a problem that may be previously explored is not plagiarism, especially given the significant improvements CLIPPER shows over ABLSim. Our results, along with our big technical differences (that even Reviewer xgWW acknowledges), suggest that our work is not a simple extension of ABLSim, but a fundamentally different approach to integrating NeSy with representation learning.

---

### Official Review · Reviewer_E58h · 2025-11-04

**Soundness:** 3
**Presentation:** 3
**Contribution:** 3
**Rating:** 6
**Confidence:** 3

**Summary:**

This work focuses on the problem of learning neurosymbolic systems where input instances must satisfy logical constraints. Traditional approaches compute all possible label combinations (pre-images) that satisfy these constraints, but this space grows exponentially. The authors propose to  prune inconsistent label combinations by using the proximity of latent representations. The key insight is that instances close in representation space are likely to share labels. The proposed pruning approach is formalized as an integer linear program that maximizes discarded pre-images while ensuring each sample retains at least one valid candidate. Experiments across benchmarks like SUM-M, MAX-M, VQAR, and MUGEN show accuracy improvements of up to 74%.

**Strengths:**

1. The paper is well-written, and the background and proposed approach are very well explained
2.  The proposed idea is well-motivated and intuitive. The challenges associated with the practical implementation of pre-image space pruning are well-discussed, and the proposed approach effectively addresses these challenges in a simple yet effective manner.
3. The evaluation sections show very promising results!
4. CLIPPER is complementary to existing NESY engines and can operate in a training-free or iterative manner.

**Weaknesses:**

1. Overhead of solving ILP is not discussed
2. Dependence on the quality of pretrained encoder or robustness to encoder noise is not discussed.

**Questions:**

1. What is the overhead of solving the proposed ILP formulation? How does it increase with the batch size?
2. How does the quality (and size) of the encoder impacts the effectiveness of CLIPPER?
3. What are the network sizes used in the work? I am curious about whether the size of the model affects the effectiveness of the proposed approach.
4. What hyperparameters does the proposed linear programming formulation introduce? What values for these hyperparameters are used in the evaluation? An ablation study would also be interesting.
5. In most of the benchmarks, the accuracy of ISED is very low, and the improvement by using CLIPPER is also minimal, can you provide more details in the evaluation section?
6. It seems from Figure 1b that the accuracy changes quite abruptly with epochs. For the final reported accuracies, how is the number of epochs determined?
7. How are the values in the evaluation tables reported? Are they the average of multiple runs? If yes, can you also include the confidence interval?

---

> ### Author Response · Authors · 2025-11-21
>
> ## On the choice of encoder
>
> Table 1 in our replies to all the reviewers shows the performance of CLIPPER w.r.t. different pre-trained and frozen encoders. Under both ResNet18 and ResNet50, the accuracy of the classifier improves substantially over the baseline in both cases. E.g., in SUM-3, n = 100, Batch size = 64, the accuracy improves from 31\% under Dolphin to 47\% and 41\% under ResNet18 and ResNet50. Please see our replies to all the reviewers for a further discussion.
>
> ## Robustness to encoder noise
> CLIPPER leads to substantial accuracy improvements in all the following scenarios:
> - Using different pre-trained and frozen encoders: ResNest18, ResNet50, and the standard MNIST classifiers CNN, Section “Robustness of CLIPPER w.r.t. encoders”.
> - Starting with a randomly initialized encoder that is being updated during NeSy learning, Section “Robustness Of CLIPPER: w.r.t. Encoders” in our replies to all reviewers, and Table 5 in the main body of our paper.
> - Using different ways to define proximity in the latent representations, see Section “Robustness Of CLIPPER: w.r.t. different notions of proximity” in our replies to all the reviewers.
>
> ## On ILP overheads
>
> Runtimes and ablations for different batch sizes are presented in our replies to all the reviewers, Section “ILP Overhead” and Table 1.
>
> ## On sizes of neural networks used
>
> We used exactly the same networks with the state-of-the-art NeSy framework for each task, see lines 724–726 in the appendix. Also, Table 1 in our replies to all reviewers shows that in SUM-$M$ and MAX-$M$, CLIPPER leads to accuracy improvements of up to 10\% (Dolphin vs Dolphin + C(MNISTNet) Random Init. & Trainable) even when using a small encoder, in particular, the CNN encoder in the standard MNIST net. We will clarify this in the paper.
>
> ## On hyperparameters
>
> The only hyperparameter is how to decide proximity in the representation space, see lines 260–262. As we state in the paragraph starting in line 176, we have different ways to define proximity graphs, in particular, to decide when two latent representations are close to each other and may be linked by an edge. We ran ablations on different ways to construct proximity graphs in our replies to all the reviewers, Section “Robustness of CLIPPER: w.r.t. different notions of proximity”.
>
> Also, we used the or-tools ILP solver and ran it with the default options, without using any hyperparameters. This can be confirmed by looking at our code – the solver is in the file `mipll_pruning_algorithm.py`. As such, running ablations for the ILP solver is not relevant to our work.
>
> ## On ISED accuracy
>
> ISED is a sampling-based NeSy approach: it samples some of the pre-images and uses only those for training. As we explain in lines 434–436, it is likely ISED prunes the gold pre-image even if CLIPPER does not prune it and passes it to ISED for training.
>
>
> ## On accuracy changes
>
> We train until the accuracy does not improve for 5 consecutive epochs. This is how the number of epochs is determined. In the case of MUGEN, we use 35 epochs, since it took around 10 hours, which we deemed sufficient for this experiment.
>
> ## About reported values
>
> For SUM-$M$, MAX-$M$, and HWF-$M$ we do three runs and report mean and standard deviation. For MUGEN, we ran it for one seed, as was done in Huang et al. (2021).

---

### Official Review · Reviewer_yU2T · 2025-11-07

**Soundness:** 3
**Presentation:** 3
**Contribution:** 3
**Rating:** 6
**Confidence:** 4

**Summary:**

The authors study the problem of learning neural classifiers whose outputs are subject to logical constraints. In this setting, the space of label combinations can grow exponentially, making learning difficult. The authors propose pruning the space of label combinations by exploiting the intuition that instances with similar latent representations are likely to share the same label

**Strengths:**

- Overall I find the paper to be quite well-written, barring the critique mentioned below

- The problem is well defined, and the solutions is derived from first principles

- The use of a running example makes it easy to follow the exposition

**Weaknesses:**

- The authors should use \citep and \citet as appropriate throghout the paper. Currently, it appears as though they only make use of \citet, which makes the paper quite a bit harder to read.

- I find lines 117-121 to be very restrictive. For instance, [1], [2], and [3] do not require that we create facts of the form digit(d, x1). Rather the characterization given is specific to logic programming, which does not encompass the entirety of NeSy.

- "Unlike supervised learning, in NeSy... the gold labels of the input instances are unknown to the learner." I only partially agree with this characterization. Most NeSy settings actually deal with semi-supervised learning where we have some seed labelled data and we have a lot of unlabelled data that we want to use. One very interesting example of this is in weakly-supervised learning [6] and [7]

- "We aim to reduce the number of candidate pre-images of the NeSy Training sampling by exploiting inconsistencies with the representation space." Isn't that addressed through semi-supervised learning using semantic loss? The exploitation of the reasoning space also bears great resemblance to NeSy entropy-regularization [6], which I expected to see as a baseline. Note that an enumeration version of this can be implemented in small domain, just to compare against the proposed approach.

- In my opinion, that is a lot of NeSy work that the authors are missing in the related works section.

- I'm not really sure how scallop and dolphin implement their losses, and how they differ semantically. Also, have the authors tried to use semantic loss as their base approach (one might achieve that through using the sampling version of semantic loss [7]). In the appendix, the authors claim that Scallops implements a scalable version of semantic loss, which I am not sure is entirely true.

- The paper could really use some figure to help illustrate the introduced concepts and methodology. It gets very hard and tedious to keep track of the added training samples in the running example.

- I found it quite a bit confusing disambiguating the use of the variable $l$ for labels as well as to range over the data points in $\mathcal{D}$. Consequently, while I intuitively understand what is meant be consistency, I failed to fully grasp example 3.3. (what does (1, x_1) -> (2, x'_1) mean?). I understand it to mean the set of non-overlapping assignments to x1 and x'_1 given that we know they are similar.

References:

[1] Jingyi Xu, Zilu Zhang, Tal Friedman, Yitao Liang, & Guy Van den Broeck. A Semantic Loss Function for Deep Learning with Symbolic Knowledge. ICML 2018.

[2] Kareem Ahmed, Stefano Teso, Kai-Wei Chang, Guy Van den Broeck, & Antonio Vergari. Semantic Probabilistic Layers for Neuro-Symbolic Learning. NeurIPS 2022.

 [3] Tao Li and Vivek Srikumar.. Augmenting Neural Networks with First-order Logic. In ACL 2019.

[4] Jessa Bekker and Jesse Davis. Learning from positive and unlabeled data: A survey. Machine
Learning 2020.

[5] Vinay Shukla, Zhe Zeng, Kareem Ahmed, & Guy Van den Broeck. A Unified Approach to Count-Based Weakly-Supervised Learning. In NeurIPS 2023.

[6] Kareem Ahmed, Eric Wang, Kai-Wei Chang, & Guy Van den Broeck. Neuro-Symbolic Entropy Regularization. In UAI 2022.

[7] Kareem Ahmed, Tao Li, Thy Ton, Quan Guo, Kai-Wei Chang, Parisa Kordjamshidi, Vivek Srikumar, Guy Van den Broeck, & Sameer Singh. (2022). PYLON: A PyTorch Framework for Learning with Constraints. NeurIPS 2021 Competitions and Demonstrations.

**Questions:**

- Regarding example 3.6, isn't the encoder mapping different instances to very close representation almost unavoidable when using neural embedding models?

- I fail to grasp the implications the absence of the optimality guarantee as stated in the first paragraph of section 3.3

- "Proposition 3.5 offers [guarantees on preserving the gold pre-images] but the formulation of problem 3.7 does not focus on that aspect." do I take that to mean that there are no such guarantee? Am I correct in my understanding that this could potentially eliminate/preclude the correct label by overpruning? Could that result in a case where a data sample is assigned a label that violates the constraint? or the set of labels is empty?

- Doesn't using a gold proximity graph in the experiments essentially solve the problem given enough data points?? A figure would be very helpful here.

- I'm very surprised by the results in Table 5. First off, why is C (Gold) worse than C (Enc)? Second, why does using Clipper with Dolphin increase the accuracy so much compared with Scallop?

- Could you say more on how your approach is used for training? Is it also used for inference? It was not clear how it is integrated into the pipeline.

---

> ### Author Response · Authors · 2025-11-21
>
> We thank the reviewer for their suggestions on appropriate citation technique and will correct this in the paper. We will also add references the reviewer feels are missing, as well as the recommended figure.
>
> ## On the problem setting
>
> We would like to note that the entire NeSy literature is huge. We are focusing on the same setting studied in DeepProbLog, Scallop, ISED, BEARS, NeuroLog, ISED, which have also been theoretically studied in Wang et al. (2023), Tsamoura et al. (2025), and Marconato et al. (2023).
>
> We see the reviewer cites the semantic loss [1]. According to Definition 1 in [1], the semantic loss is the cross entropy of the weighted model counting (WMC) of a **propositional formula.** ProbLog (and DeepProbLog, Scallop that are based on the ProbLog semantics) utilize WMC to compute the probabilities the derived facts are true under the possible world semantics, “Probabilistic Programming Concepts”, 2015. ProbLog does not deal with propositional but with relational theories. So, ProbLog reduces to WMC via propositionalization.
>
> Based on the above, we do not see the setting we study as more restrictive than [1] and [6] (we can see that [6] is again based on WMC and propositional theories).
>
> ## On the level of supervision in NeSy settings
>
> As we wrote above, we focus on the setting studied in DeepProbLog, Scallop, ISED, BEARS, NeuroLog, ISED, which have also been theoretically studied in Wang et al. (2023), Tsamoura et al. (2025), and Marconato et al. (2023). All these settings assume that no supervised samples are available in the general case. So, we don’t quite agree with the reviewer’s comment that “most NeSy settings actually deal with semi-supervised learning where we have some seed labelled data and we have a lot of unlabelled data that we want to use”.Of course, our setting is an umbrella of different weakly supervised settings as also discussed in Wang et al. (2023) and Tsamoura et al. (2025). We will clarify the above in a revised version of our work.
>
> ## Clarifying the reduction of pre-images in CLIPPER
>
> CLIPPER aims to reduce the number of pre-images, providing “stronger” supervision during training. As we discuss in the lines 152 – 160, theoretical and empirical studies have shown that a reduction in the number of pre-images per training sample leads to classifiers with higher accuracy. For example, Tsamoura et al. (2025) showed that the probability a classifier misclassifies instances of the given class is a direct function of the number of pre-images. Marconato et al. (2023) offers another theoretical analysis along these lines. Semantic loss or the work in [6] do not reduce the number of pre-images and we see this research orthogonal to our work.
>
> Notice also that Scallop relies on approximations of semantic loss (semantic loss is essentially WMC with a cross entropy term as we discuss above). Our empirical results show that CLIPPER brings substantial improvements when combined with semantic loss.
>
> ## On the losses for Scallop and Dolphin
>
> Scallop offers a top-k approximation to WMC, Section 4.1 in Huang et al. (2021). SL is the cross entropy of WMC (Definition 1 in [1]). Hence, we see Scallop as a scalable version of SL.
> We tried using SL directly, however, for scalability reasons and due to the large number of pre-images in our scenarios (see our replies to Reviewer jDFH that shows the number of mean pre-images per scenario) this was not possible. Notice that computing the WMC is one issue; computing the pre-images is also another difficult problem for which Scallop and Dolphin offer a scalable implementation.
> In the case of Dolphin, we use fuzzy semantics via the DAMP provenance, as it was empirically shown to be the better provenance in their paper for the benchmarks being considered.

---

> > ### Author Response · Authors · 2025-11-21
> >
> > ## On guarantees about the gold pre-images
> >
> > We cannot guarantee that the gold pre-images are maintained, as there may exist edges between instances of different classes, resulting in a pruning that discards the gold pre-image. As discussed in Section 3.1, these cases may occur when the encoder maps instances of different classes very close in the representation space and access to the gold pre-images is missing.
> >
> > As we discuss in our replies to all the reviewers, Section “Pruning And Gold Pre-Images”, our formulation (Problem 3.7 and Proposition 3.8) ensures that:
> > - No NeSy sample is associated with zero pre-images (the notion of soundness, Definition 3.4) after pruning. Hence, samples with a single pre-image only, e.g., supervised samples, are guaranteed to keep their single and gold pre-image.
> > - We can guarantee that the gold pre-images are maintained ONLY when we prune w.r.t. the gold proximity graph, see Proposition 3.5.
> > - In specific scenarios, we **can guarantee that the gold pre-images are maintained.** One such case is when the gold proximity graph is undirected and for each instance in a NeSy sample, there exists a path to an instance $x^*$ associated with a single (and gold due to the assumption that each NeSy sample is associated with all the pre-images that adhere to the constraints, including the gold one) pre-image only. Then, the gold pre-image of each training sample will be maintained. Please see our replies to all the reviewers, Section “Pruning And Gold Pre-Images” for a proof sketch.
> >
> > So while gold labels can potentially be eliminated by overpruning, as we show in our replies to all reviewers, Section "Pruning And Gold Pre-Images” and Table 1:
> > - When the encoder is pre-trained and fixed, CLIPPER is quite effective in maintaining the gold pre-images, maintaining, on average, from the 88\% (worst case) to 98\% (best case) of the gold ones.
> > - When the encoder is trainable, CLIPPER is quite effective in maintaining the gold pre-images, maintaining, on average, from the 94\% (worst case) to 98\% (best case) of the gold ones, row “+ CLIPPER(MNISTNet) Random Init. & Trainable” in Table 1”.
> >
> > ## Could that result in a case where a data sample is assigned a label that violates the constraint or the set of labels is empty?
> >
> > No. Our formulation (Problem 3.7 and Proposition 3.8) ensures that:
> > - The retained pre-images adhere to the constraints across different training samples (the notion of consistency, Definition 3.2). The notion of consistency is demonstrated in Example 3.3.
> > - No NeSy sample is associated with zero pre-images (the notion of soundness, Definition 3.4) after pruning. Hence, samples with a single pre-image only, e.g., supervised samples, are guaranteed to keep their single and gold pre-image.
> >
> > ## Doesn't using a gold proximity graph in the experiments essentially solve the problem given enough data points??
> > As we stated earlier and similarly to DeepProbLog, Scallop, ISED, NeuroLog, and ISED, we focus on the setting where we do not have access to the gold labels. Our setting has also been theoretically studied in Wang et al. (2023), Tsamoura et al. (2025), and Marconato et al. (2023). The absence of the gold labels implies that we do not have access to the gold proximity graph either.
> >
> > ## Why is C (Gold) worse than C (Enc)
> > As we state in lines 405-406, in MUGEN, this is because we did not have access to the gold labels; we approximated C(GOLD) using a pretrained encoder. So, we essentially pruned with labels that might be noisy and not the actual gold ones. Also, as we state in lines 443–447, “For MUGEN, the accuracy of the baseline DOLPHIN model increases from 33.9% to 84% for n = 500 under pruning guided by the gold proximity graph. Instead, when a non-gold proximity graph is employed, the accuracy for the same scenario increases from 33.9% to 86.6%. This is because, in MUGEN, we approximated C(GOLD) using a pretrained encoder, i.e., the labels that we use to compute the gold proximity graph are noisy.”
> >
> > ## Second, why does using Clipper with Dolphin increase the accuracy so much compared with Scallop?
> > For all benchmarks, we use the implementations provided by Scallop and Dolphin and their respective source codes. There are cases where the symbolic program differs between Scallop and Dolphin for HWF and MUGEN, which explains the difference in accuracies in our limited data settings.
> >
> > ## On using CLIPPER for Training
> >
> > To support this scenario, the outputs of the encoder must be connected to the output classification layer. So, we can use the encoder to decide proximity in the representation space, and, hence, define the ILP in Eqn 1 and then during NeSy training, we can backpropagate through the encoder to update its weights.

---

> ### Author Response · Authors · 2025-11-21
>
> ## On using CLIPPER for Inference
>
> As we state in lines 121–122, different NeSy frameworks may employ different reasoning semantics at testing time which is orthogonal to this work.
> In our paper, we use CLIPPER at training time. However, using CLIPPER at testing time is definitely possible. We can essentially use CLIPPER to prune pre-images and then, instead of training using the remaining pre-images, we can use them for inference, e.g., compute the WMC under the ProbLog semantics.
>
> ## On integrating CLIPPER into the pipeline
>
> We show the steps of CLIPPER in Algorithm 1. In brief, the pipeline is as follows:
> - We collect the pre-images (e.g., Scallop, Dolphin) or a sample of all pre-images (ISED)
> - We prune the pre-images using Algorithm 1
> - We use the remaining pre-images for training, e.g., using the top-k WMC of the pre-images under Scallop. As we state in line  147, our work is orthogonal to the actual loss used for training.

---

### Official Review · Reviewer_Mwcy · 2025-11-08

**Soundness:** 2
**Presentation:** 2
**Contribution:** 2
**Rating:** 2
**Confidence:** 4

**Summary:**

This paper proposes a method for improving neurosymbolic learning (NESY) by pruning inconsistent label combinations through representation-space similarity. Traditional NESY systems must handle exponentially many label combinations that satisfy logical constraints, making training inefficient and ambiguous. CLIPPER leverages the intuition that instances close in latent space likely share labels. The method constructs proximity graphs among instances based on encoder similarity, defines consistency constraints for pre-images (label combinations), and formulates an integer linear program (ILP) that discards inconsistent pre-images while preserving at least one valid combination per training sample. Experiments across multiple NESY engines and benchmarks (SUM-M, MAX-M, HWF-7, VQAR, and MUGEN) demonstrate large performance gains, suggesting that pruning inconsistent label configurations improves learning efficiency and disambiguation.

**Strengths:**

- The proposed method addresses the exponential explosion of label combinations in NESY with an elegant pruning approach.
- The authors provide clear definitions, soundness guarantees, and ILP optimality proof.

**Weaknesses:**

- The method leverages standard ideas (representation similarity + graph consistency + ILP) without major theoretical innovation.
- ILP optimization over large NESY datasets may be computationally expensive; no runtime or complexity analysis is reported.
- The effectiveness heavily depends on the quality of latent representations; the paper does not explore failure cases or encoder ablation.
- While numerical gains are large, the work lacks intuitive analysis or visualization of what pre-images were pruned and why.
- Heavy formalism and overloaded notation could be streamlined for clarity.

**Questions:**

- How scalable is CLIPPER to large NESY datasets when solving the ILP on full mini-batches? Is there an approximate or relaxed formulation (e.g., LP relaxation or greedy pruning)?
- How sensitive is performance to encoder choice (e.g., random vs. pretrained vs. jointly trained)?
- Does pruning risk discarding gold pre-images under imperfect proximity graphs? Are there empirical checks or safety mechanisms?
- Could the method generalize to multi-modal encoders or symbolic rules with uncertainty (e.g., probabilistic logic)?
- How does CLIPPER interact with end-to-end differentiable reasoning frameworks like DeepProbLog or Neural Module Networks?

**Details Of Ethics Concerns:**

No ethics concerns.

---

> ### Author Response · Authors · 2025-11-21
>
> ## On the innovations of our contribution
> Thank you for your comment.
> + Our primary objective is to leverage the power of latent representations to enhance NeSy learning rather than to develop better techniques for learning encoders. As we claim in lines 84–86 and similar to He et al. (2024), CLIPPER is designed to be employed in a training-free fashion if needed, i.e., without simultaneously training the underlying encoder. Of course, our empirical results show that the underlying encoder also has higher accuracy when simultaneously trained with the classifier (experiments on MUGEN, Table 5 and Table 1 in our replies to all the reviewers).
> + CLIPPER is far from being straightforward to develop. One close competitor to our work, ABLSim (NeurIPS 2021) – which we were made aware of by one of the reviewers –  supports only a specific NeSy setting, ABL, that trains using a single pre-image only (Bridging Machine Learning and Logical Reasoning by Abductive Learning, NeurIPS 2019). In addition, ABLSim, being prone to local minima as ABL, suffers from very low accuracy when the number of pre-images per NeSy sample increases. Please see our detailed responses to Reviewer xgWW for an analysis of ABL and ABLSim.
> + We already proved several properties of our formulation, see Section 3.1, Proposition 3.5, Proposition 3.8, and Problem 3.7.
> + However, as we are intrigued by your comment, can you please give an indication of the theoretical results that you would like us to prove and provide us with references to relevant literature to see how these results could be extended/adapted to our setting?
>
> ## About the cost of ILP solving
> We present new ablations in our replies to all reviewers. The conclusions that we can draw are that CLIPPER leads to substantial accuracy improvements over the baseline (1) under different pretrained and frozen encoders, see Table 1, Section “Robustness of CLIPPER: w.r.t. encoders”, (2) when using randomly initialized and trainable encoders, Section “Robustness Of CLIPPER: w.r.t. encoders”, and (3) under different notions for defining proximity in the representation space, Section “Robustness of CLIPPER w.r.t. different notions of proximity”.
>
> ## What preimages were pruned?
> We added several examples demonstrating the intuition of our technique, including Example 1.1 and 2.1 that present the high-level intuition. About showing why certain pre-images are pruned in our experiments: Given that the batch sizes are much greater than one and that the number of pre-images is large, the number of equations in the ILP formulation in Eqn (1) – the solution of which explains why certain pre-images are preserved while others are pruned –  will become quite large and hence, it will be difficult to parse.
>
> If the reviewer wishes, we can provide the exact ILP formulation and its solution for one of our simplest scenarios, e.g., SUM-3, we are happy to do so.
>
> ## On the formalism and notations
> We point the reviewer to Table 6 in the appendix that summarizes our notation. Please let us know of any specific suggestions to improve our notation.
>
> ## On scalability to large datasets
> Regarding scalability and runtime for different mini-batches, please see our reply to all the reviewers, Section “ILP Overhead”.
> As we note in lines 230-241, greedy pruning may lead to unsound prunings.
> We can do clever batching of the samples to maximize the effects of pruning as we did in VQAR by grouping in the same batch samples with few pre-images (e.g., samples with one pre-image only) and samples with many pre-images.
>
> ## On encoder choice
> Please see our replies to all reviewers, Section “Robustness of CLIPPER: w.r.t. encoders”.
> Starting with a randomly initialized encoder that is not trainable does not make much sense to experiment with, as this setting is equivalent to randomly pruning pre-images.
>
>
> ## On the risk of discarding gold pre-images
> Please see our responses to all the reviewers, Section “Pruning and gold pre-images” for a discussion on this and further empirical results. In brief:
> + Supervised samples are guaranteed to keep their single and gold pre-image.
> + We can guarantee that the gold pre-images are maintained ONLY when we prune w.r.t. the gold proximity graph, see Proposition 3.5. The gold proximity graph is UNKNOWN since we assume that we do not have access to the gold pre-images.
> + The retained pre-images adhere to the constraints across different training samples (the notion of consistency, Definition 3.2).
> + In specific scenarios, we **can guarantee that the gold pre-images are maintained.** One such case is when the gold proximity graph is undirected and for each instance in a NeSy sample, there exists a path to an instance $x^*$ associated with a single pre-image only.
>
> The retained preimages adhere to these constraints. Please let us know if you have specific safety mechanisms in mind.

---

> > ### Author Response · Authors · 2025-11-21
> >
> > ## On the generalizability of the method
> >
> > **About symbolic rules with uncertainty:** As we state in line 121—122, we support different frameworks that may employ different reasoning semantics at testing time. This is possible by abstracting the background knowledge $\mathcal{K}$ and $\phi$ via abduction as stated in lines 135—137. Scallop, in particular, is a successor of DeepProbLog and it is based on the ProbLog semantics.
> >
> > **About generalizing to multiple encoders:** The answer is yes. The only change we need to do is to use different encoders for deciding when two instances are close in the representation space, lines 259—261, e.g., instead of using a single $h$ to use multiple ones depending on the instances. As stated in lines 176—184, our work is quite flexible, allowing users to support any distance measure and encoder they prefer.
> >
> > ## On interaction with end-to-end frameworks
> >
> > Scallop is a scalable successor of DeepProbLog that has shown better results in their paper, so it is reasonable to expect that CLIPPER will interact the same way with DeepProbLog as it does with Scallop and Dolphin. In fact, Scallop, ISED and Dolphin are end-to-end differentiable reasoning frameworks, and have SOTA performance.

---

### Author Response · Authors · 2025-11-21
**Response to All Reviewers**

We thank the reviewers for their feedback and suggestions, and will work on incorporating them in our revised paper. We have added all the tables mentioned in our reply to the revised paper, and will continue updating results for ongoing experiments. All revised text in our paper will be highlighted in blue.

## ILP overhead
In Table 1 below, we report the average time to solve the ILP in Eqn 1 and pruning the pre-images per batch for SUM-3 and SUM-4 under different batch sizes and encoders. We add this table to our Appendix D (Tables 7 and 8). Our empirical analysis shows that
**the runtime of CLIPPER is small despite the fact that the number of pre-images is quite large.**

Table 1: Ablations on SUM-3 and SUM-4 for different encoders, batch sizes, and frozen & pre-trained vs randomly initialized and trainable encoders. In the rows “+ C(ResNet18) Pretrained & Frozen” and  “+ C(ResNet50) Pretrained & Frozen”, the encoder is pretrained and frozen. In the row “+ C(MNISTNet) Trainable”, the encoder is randomly initialized and trainable. MNISTNet is the CNN encoder used in the standard MNIST classifier. NA stands for non-applicable. Times are in seconds.
Experiment |Algorithm | Classification Accuracy | % Retained Preimages | % Preimages with Ground Truth | Avg Pruning Time (s) | Avg Time to Solve ILP (s)
-|-|-|-|-|-|-
|SUM-3, n = 100, Batch size = 64 |DOLPHIN | 31.6 | NA | NA | NA | NA
| |\+ C(ResNet18) Pretrained & Frozen   | 47.09 | 77.99 | 91.56 | 0.49 | 0.03
| |\+ C(ResNet50) Pretrained & Frozen | 41.89 | 75.7 | 88.38 | 0.51 | 0.03
| |\+ C(MNISTNet) Trainable | 41.51 | 78.12 | 94.81 | 0.53 | 0.03
|SUM-3, n = 100, Batch size = 128|DOLPHIN | 32.86 | NA | NA | NA | NA
||\+ C(ResNet18) Pretrained & Frozen   | 49.24  | 80.17  | 92  | 1.88  | 0.06
||\+ C(ResNet50) Pretrained & Frozen | 42.56  | 74.21  | 90  | 1.5  | 0.07
||\+ C(MNISTNet) Trainable | 36.84  | 77.85  | 96.21  | 1.39  | 0.07
|SUM-4, n = 100, Batch size = 64|DOLPHIN | 31.43  | NA | NA | NA | NA
||\+ C(ResNet18) Pretrained & Frozen   | 31.28  | 87.5 | 96.19 | 3.39 | 0.39
||\+ C(ResNet50) Pretrained & Frozen | 28.69  | 86.46 | 92.9 | 3.42 | 0.42
||\+ C(MNISTNet) Trainable | 35.19  | 85.88 | 98.58 | 3.34 | 0.39

Regarding the complexity of our ILP formulation, notice the following:
1. Without any approximations, the complexity of the SOTA NeSy loss, Semantic Loss Xu et al. (2018), is #P-complete (Mark Chavira and Adnan Darwiche. On probabilistic inference by weighted model counting. Artificial Intelligence, 2008). In contrast, ILP is an NP-hard problem.
2. There are multiple heuristics to solve ILP efficiently, e.g., “Local Branching Relaxation Heuristics for ILP” (Huang et al., 2023).
3. We can do clever batching of the samples to maximize the effects of pruning as we did in VQAR. In particular, we can group in the same batch samples with very few pre-images (e.g., samples with one pre-image only) and samples with many pre-images. Due to (1) the property of soundness (Definition 3.4), (2) our problem formulation (Problem 3.7) and (3) the properties of our ILP formulation (Proposition 3.8), the pre-images of the former NeSy samples will be maintained, forcing the pre-images of the latter NeSy samples to be pruned. We will elaborate on this in the revised version of our work.
4. As we explain in lines 230–241, naive greedy pruning compromises soundness.

Since our work focuses on a plug-and-play approach to improve NeSy learning using latent representations, we consider the choice of ILP solver to be orthogonal.

---

> ### Author Response · Authors · 2025-11-21
>
> ## Pruning and gold pre-images
>
> We start by providing additional statistics on the effectiveness of CLIPPER in pruning non-gold pre-images. Table 1 from our earlier comment shows that:
> + CLIPPER is quite effective in reducing the number of pre-images, pruning, on average, up to 25\% of pre-images (Column “Percentage of retained pre-images”).
> + When the encoder is pre-trained and fixed, CLIPPER is quite effective in maintaining the gold pre-images, maintaining, on average, from the 88\% (worst case) to 98\% (best case) of the gold ones.
> + When the encoder is trainable, CLIPPER is quite effective in maintaining the gold pre-images, maintaining, on average, from the 94\% (worst case) to 98\% (best case) of the gold ones, row “+ C(MNISTNet) Trainable” in Table 1”.
>
> Second, notice that our formulation (Problem 3.7 and Proposition 3.8) ensures that:
> + No NeSy sample is associated with zero pre-images (the notion of soundness, Definition 3.4) after pruning.
> + Due to the above property, samples with a single pre-image only, e.g., supervised samples, are guaranteed to keep their single and gold pre-image.
> + The retained pre-images adhere to the constraints across different training samples (the notion of consistency, Definition 3.2). The notion of consistency is demonstrated in Example 3.3.
> + We can guarantee that the gold pre-images are maintained ONLY when we prune w.r.t. the gold proximity graph, see Proposition 3.5. The gold proximity graph is UNKNOWN since we assume that we do not have access to the gold pre-images – corner cases in which some samples are supervised or have a single pre-image may exist and are supported by our formulation.
> + We cannot guarantee that the gold pre-images are maintained, as there may exist edges between instances of different classes, resulting in a pruning that discards the gold pre-image. As discussed in Section 3.1, these cases may occur when the encoder maps instances of different classes very close in the representation space and access to the gold pre-images is missing.
>
> In specific scenarios, we **can guarantee that the gold pre-images are maintained.** One such case is when the gold proximity graph is undirected and for each instance in a NeSy sample, there exists a path to an instance $x^*$ associated with a single (and gold due to the assumption that each NeSy sample is associated with all the pre-images that adhere to the constraints, including the gold one) pre-image only. Then, the gold pre-image of each training sample will be maintained. We can prove this inductively:
> + In the base case, the single (and gold) pre-image of associated with $x^\*$ will be maintained due to the notion of soundness, Definition 3.4, our formulation (Problem 3.7), and Proposition 3.8.
> + The gold pre-images associated with each instance $x$ that is connected with $x^\*$ will be maintained since (1) $x^\*$ is associated with the gold pre-image and (2) the edge between $x$ and $x^\*$ is consistent with gold pre-image and hence all pre-images that include the gold label will be maintained (pruning (Definition 3.4) eliminates all pre-images that are inconsistent with edges only).
> + Similarly, the gold pre-images of instances that have a path with $x^\*$ via $x$ will be maintained via exactly the same reasoning.
>
> ## Robustness of CLIPPER
>
> Table 1 shows the performance of CLIPPER w.r.t. different pre-trained and frozen encoders. We see that, under both ResNet18 and ResNet50, the accuracy of the classifier improves substantially over the baseline in both cases. E.g., in SUM-3, n = 100, Batch size = 64, the accuracy improves from 31\% under Dolphin to 47\% and 41\% under ResNet18 and ResNet50.
>
> ### With respect to encoders
> Results on randomly initialized and trainable encoders are shown in Table 1:
> + We see that for pretrained encoders such as ResNet18 and ResNet50, the accuracy of the classifier improves substantially over the baseline in both cases. E.g., in SUM-3, n = 100, Batch size = 64, the accuracy improves from 31\% under Dolphin to 47\% and 41\% under ResNet18 and ResNet50.
> + Table 1 also shows that one can use the trainable model itself as the encoder. E.g., in SUM-3, n = 100, Batch size = 64, the accuracy of the baseline model improves from 31\% to 41\% when pruning subject to a standard MNIST CNN that has been randomly initialized and whose weights are being updated during learning.
> + In MUGEN (the experiments where we have performance gains of 50\%, Table 5), we start with a *randomly initialized* encoder and interleave pruning with learning of the encoder’s weights, i.e., during training, we use the encoder to prune pre-images, backpropagating through it at the same time.

---

> ### Author Response · Authors · 2025-11-21
>
> ### With respect to different notions of proximity
>
> We assess the robustness of CLIPPER w.r.t. different approaches for determining proximity in the representation space, as shown in lines 176–184. Table 2 reports results on DOLPHIN + CLIPPER when using a different way to define proximity between latent representations. In particular, for each instance $x$, we only consider the 5\% and the 10\% closest instances, while constructing the ILP. The accuracy of the baseline Dolphin model for the corresponding SUM-3 and SUM-4 scenarios is 31\% and 32\%, respectively. The above illustrates the robustness of CLIPPER w.r.t. different notions of proximity. In our experiments, the results were obtained by considering a top-$k$ proximity.
>
> Table 2: Accuracy of Dolphin + C(ResNet18)
>
> | Experiment | Proximity (5%) | Proximity (10%) |
> |-|-|-|
> SUM-3, n = 100, Batch size = 64 | 42.3 (Retrained pre-images 84.5%, Retrained pre-images with GT 90.45%) | 33.61 (Retrained pre-images 84.16%, Retrained pre-images with GT 90.41%)
> SUM-4, n = 100, Batch size = 64 | 33.4 (Retrained pre-images 97.59, Retrained pre-images with 98.49) | 39.45 (Retrained pre-images 94.43, Retrained pre-images with 96.33)
>
> ## About plagiarism (Reviewer xgWW)
> We saw that the Reviewer xgWW in their comments flagged our work for ethics review: “Flag For Ethics Review: Yes, Research integrity issues (e.g., plagiarism, dual submission)”. The technique we supposedly copied is ABLSim, “Huang et al., Fast Abductive Learning by Similarity-based Consistency Optimization. NeurIPS 2021.”
>
> At the time of writing our paper, in spite of our extensive literature survey, we did not encounter ABLSim until the reviewer pointed it out to us. However, we are willing to cite ABLSim and compare against it as a baseline (see the tables below).
>
> Having said that, we strongly protest the allegations of plagiarism and consider them to be untrue and unethical. We list our reasons as follows:
> * ABLSim is an adhoc technique for NeSy learning. Unlike ABLSim, CLIPPER can be plugged on top of several frameworks, such as Scallop, Dolphin and ISED, leading to substantially higher classification accuracy over the baselines.
> * ABLSim’s consistency measure (Eq. 12 in Huang et al., NeurIPS 2021) is different from our notion of consistency, see Def 3.2 in our paper.
> * We formalize under which cases the gold pre-images can be maintained, Proposition 3.5 in our paper. Further details about pruning gold pre-images under CLIPPER can be found in our replies to all Reviewers, Section “Pruning and gold pre-images”. ABLSim gives NO such guarantees.
> * Our problem formulation, Problem 3.7 in our paper, and the solution to our problem, Eqn 1 in our paper, is very different from ABLSim’s objective, Eq. 12 in Huang et al., NeurIPS 2021.
>
> These are *fundamental* differences in our contributions, and as such, CLIPPER cannot be considered an extension of the ABLSim technique. **We therefore request the reviewer to retract their claims about plagiarism.**
>
> Finally, ABLSim is an extension of ABL “Dai et al., Bridging Machine Learning and Logical Reasoning by Abductive Learning. NeurIPS 2019.” As previous research has shown, see Tsamoura et al. (2021) and Wang et al. (2023), ABL is prone to local minima leading to very low classification accuracy compared to other baselines such as DeepProbLog and Scallop. The experimental results of Tsamoura et al. (2021) and Wang et al. (2023) offer a detailed discussion.
>
> We ran ABLSim for MNIST Sum-3, Sum-4, and Max-3 for 100 training samples. The table below shows the results that we obtained after running ABLSim on our scenarios for 100 epochs. We can see that ABLSim suffers from the same limitations as AB, leading to classification accuracies lower than just the plain baseline frameworks.
>
> Table 3: Comparison of classification accuracies of Dolphin and Dolphin + CLIPPER with ABLSim. Batch size used was 64.
> | Algorithm | Sum-3, n=100 | Sum-4, n=100 |
> |-|-|-|
> |Dolphin| 31.6 | 31.43 |
> | \+ C(ResNet18) Pretrained & Frozen | 47.09 | 31.28 |
> | \+ C(MNISTNet) Trainable | 41.51 | 35.19 |
> | ABLSim | 16.51 | 11.71 |
>
> We have added these results for ABLSim in the paper (Tables 1 and 2 in our revised paper) and will continue our experiments on ABLSim for SUM-$M$, MAX-$M$, and HWF-7.
>
> Finally, for MAX-3, we obtained the following results: Dolphin:  61.86 ± 2.52 (Table 2 in our paper), Dolphin + CLIPPER(ResNet18) Pretrained & Frozen: 63.57 ± 3.41 (Table 2 in our paper), ABLSim: 42.48. Again, we can see that ABLSim, instead of improving, leads to worse accuracy than the baseline, an observation also made in Tsamoura et al. (2021) and Wang et al. (2023).
>
> As you can see, CLIPPER leads to substantially higher accuracy than ABLSim. These results can only be obtained by fundamental insights into representation-driven pruning strategies and not via simply extending ABLSim.

---

### Author Response · Authors · 2025-12-03
**A Summary of our Changes**

We thank all the reviewers for their insightful comments and suggestions. We appreciate the positive remarks regarding our approach and evaluation results.

At the suggestion of some reviewers, we have added experimental results detailing the fraction of retained pre-images and the fraction of those containing gold labels. We also show the amount of time taken to prune these preimages. These results are added in Tables 1 and 2 in the experiments section of our revised paper, and in Tables 7, 8, and 9 in Appendix D. We also show the accuracy curves for MUGEN trained on the full dataset in Figure 3, Appendix D. We also add ablations regarding the complexity of the encoder used for CLIPPER, as well as the batch size.

At reviewer xgWW’s suggestion, we additionally cite ABLSim and ABLKit as two baselines, and are actively conducting experiments for them over relevant benchmarks. Preliminary results using these baselines can be found in Tables 1 and 2 of the revised paper.

---

### Meta-Review · Area_Chair_Gwi6 · 2026-01-02

**Summary:**

**Summary of contribution** \
The paper addresses the symbol grounding problem in neuro-symbolic (NeSy) systems. The central idea is to exploit representation similarity derived from an auxiliary encoder to mitigate label ambiguity originating from the symbolic component. This is achieved by formulating an integer linear programming problem whose optimal intermediate labels are jointly consistent with the similarity graph induced by the encoder and the labels inferred by the symbolic reasoner.

**Summary of concerns** \
Two major concerns remain unaddressed following the rebuttal:
1. **Novelty** Several important related works were missing from the original submission, as noted by three reviewers (Mwcy, yU2T, xgWW). While the authors have revised the manuscript to include some of the missing references, significant prior work remains unacknowledged. In particular, the use of learned representations in NeSy was first introduced in [1] via autoencoders. Subsequent work in [2], and its improved version in [3], demonstrated how self-supervision and representation learning can be leveraged in a principled manner to reduce label ambiguity in low-data and weakly supervised NeSy settings. The current paper lacks a discussion of, and comparison with, these closely related approaches, making it difficult to clearly assess its novelty and positioning.
2. **Quality** The experimental evaluation is incomplete in several respects. First, important baselines such as ABSLim and GEDI are partially or entirely missing. Second, as highlighted by reviewer a97x, the experiments deviate from standard practice by focusing exclusively on an extreme low-data regime and on a selected subset of samples for worst-case analysis. To provide a comprehensive evaluation and to clearly demonstrate the advantages of the proposed approach, the experimental analysis should be extended to include standard NeSy evaluation settings and a more complete set of relevant baselines.

**Decision** \
The paper proposes an interesting and technically sound approach to addressing the symbol grounding problem in NeSy systems. All reviewers recognize the importance of the problem and acknowledge the validity of the proposed solution. However, in its current form, the paper is incomplete. Substantial revisions are required to better position the work with respect to existing literature in NeSy and to strengthen the experimental evaluation by incorporating missing baselines and adhering to standard evaluation practices.

**Additional references** \
[1] Misino, Marra, Sansone. VAEL: Bridging Variational Autoencoders and Probabilistic Logic Programming. NeurIPS 2022 \
[2] Sansone, Manhaeve. Learning Symbolic Representations Through Joint GEnerative and DIscriminative Training. ICLR Workshop 2023 \
[3] Sansone, Manhaeve. Unifying Self-Supervised Clustering and Energy-Based Models. TMLR 2025

**Reviewer Concerns:**

**Concerns (in addition to the ones mentioned above)** \
1. The proposed approach relies on a strong and potentially limiting assumption (reviewers Mwcy, E58h, xgWW, and jDFH) namely, the availability of a high-quality encoder. While the authors present preliminary experiments using different pre-trained models, this concern remains largely unaddressed. In particular, it is unclear how robust the method is when this assumption is weakened.
2. The proposed solution introduces additional computational overhead, as noted by reviewers Mwcy, E58h, and jDFH. Although the authors include an initial analysis of the computational cost, this evaluation is restricted to the limited experimental setting discussed above. Consequently, it remains unclear whether the increased computational burden is justified by corresponding performance gains in more standard evaluation scenarios and in larger-scale data regimes.

**Reviewer Scores:**

Reviewers would have kept their score, as the paper is still incomplete after the rebuttal.

---

### Decision · Program_Chairs · 2026-01-26

Reject